# SHARP: Accelerating Language Model Inference by SHaring Adjacent layers with Recovery Parameters

## Abstract

While Large language models (LLMs) have advanced natural language processing tasks, their growing computational and memory demands make deployment on resource-constrained devices like mobile phones increasingly challenging. In this paper, we propose SHARP (SHaring Adjacent Layers with Recovery Parameters), a novel approach to accelerate LLM inference by sharing parameters across adjacent layers, thus reducing memory load overhead, while introducing low-rank recovery parameters to maintain performance. Inspired by observations that consecutive layers have similar outputs, SHARP employs a two-stage recovery process: Single Layer Warmup (SLW), and Supervised Fine-Tuning (SFT). The SLW stage aligns the outputs of the shared layers using $\mathcal{L}_2$ loss, providing a good initialization for the following SFT stage to further restore the model performance. Extensive experiments demonstrate that SHARP can recover the model's perplexity on various in-distribution tasks using no more than 50k fine-tuning data while reducing the number of stored MLP parameters by 38% to 65%. We also conduct several ablation studies of SHARP and show that replacing layers towards the later parts of the model yields better performance retention, and that different recovery parameterizations perform similarly when parameter counts are matched. Furthermore, SHARP saves 42.8% in model storage and reduces the total inference time by 42.2% compared to the original Llama2-7b model on mobile devices. Our results highlight SHARP as an efficient solution for reducing inference costs in deploying LLMs without the need for pretraining-scale resources.

## 1 Introduction

Following the principles of scaling laws, large language models (LLMs) have become one of the central topics in Natural Language Processing (NLP) (Brown, 2020; Zhang et al., 2022; Hoffmann et al., 2022; Bubeck et al., 2023; Chowdhery et al., 2023; Bai et al., 2023; Team et al., 2023; Touvron et al., 2023). However, deploying a pre-trained large language model requires significant computational and memory resources (Aminabadi et al., 2022; Pope et al., 2023; Kim et al., 2023b; Zhang et al., 2024b), which may further restrict their inference speed. For instance, a 70-billion-parameter language model stored in FP16 precision requires approximately 148GB of memory to hold the model weights, necessitating two A100 GPUs with 80GB of memory each to load the entire model. During inference, the entire input sequence and the KV cache are also stored on the GPU, incurring additional memory usage. Although techniques like layer-wise inference (HuggingFace, 2022), which load the model to GPU layer by layer, enable LLM inference on a single GPU, they introduce additional inference latency due to frequent memory loading or disk reading. In particular, these concerns are significant for deployment on mobile devices, which typically have smaller DRAM (e.g., around 6GB in the iPhone 15) and higher communication overhead (Liu et al., 2024).

To alleviate these issues, several methods have been carefully explored. One direction is to optimize the calculation process of the attention mechanism and the storage of the KV cache, including Reformer (Kitaev et al., 2020), Flash Attention (Dao et al., 2022; Dao, 2023; Shah et al., 2024), $H_2O$ (Zhang et al., 2024b), and so on. Another main direction is to compress the existing model while retaining model performance, including quantization (Dettmers et al., 2022; Liu et al., 2023a; Kang et al., 2024), pruning (?Sun et al., 2023), and sparsification (Frantar & Alistarh, 2023; Dong &

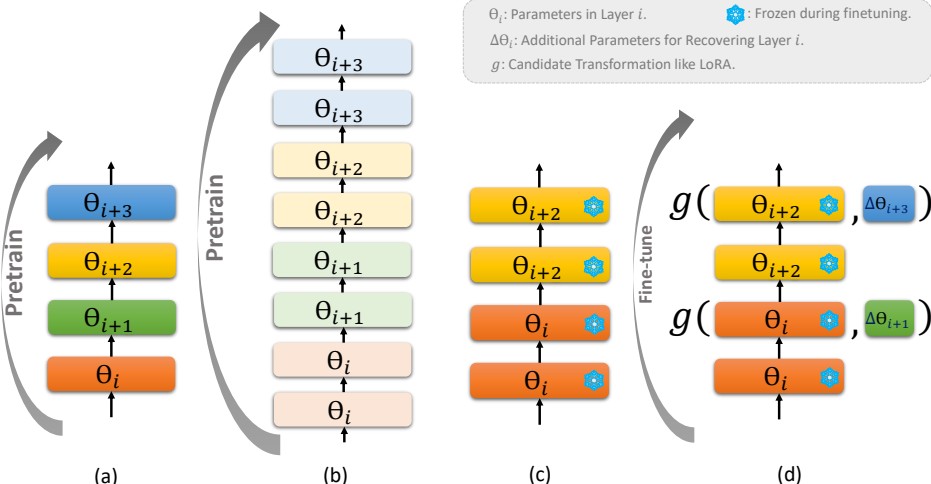

Figure 1: (a) Regular pretrained baseline model without layer sharing. (b) Adjacent layer sharing used in MobileLLM (Liu et al., 2024). They repeat the layer twice and train the model from scratch. (c) Direct Sharing: directly apply vanilla adjacent layer sharing to the pretrained model to accelerate inference. (d) **(Ours) SHARP**: SHaring Adjacent Layers with Recovery Parameters. SHARP leverages fine-tuning-scale data to train additional parameters $\Delta\Theta$, which consist of far fewer parameters than the original $\Theta$, in order to recover the model's performance. In this paper, we explore several candidate transformations, including the LoRA-style function, to apply the additional parameters

Chen, 2024; Mirzadeh et al., 2023; Song et al., 2024). There are also other methods that try to accelerate inference by optimizing decoding algorithms, such as speculative decoding (Kim et al., 2024). Additionally, it's worth mentioning some other research also tries to directly train small language models (SLMs) (Black et al., 2022; Zhang et al., 2022; Timiryasov & Tastet, 2023; Dey et al., 2023; Biderman et al., 2023; Gunasekar et al., 2023; Hu et al., 2024) rather than compressing existing large language models. However, this always requires pretraining-scale resources, which cost more than the methods that only use post-training-scale data for recovery, such as sparsification.

In this paper, we focus on a new methodology for efficient inference on current pretrained models, named the *adjacent layer-sharing strategy*. It is partially inspired by the observation from Deja Vu (Liu et al., 2023b): in Figure 5(a) and (b) of their paper, they show that the cosine similarity between representations at two consecutive layers, or even a few layers apart, can be very high (greater than 95%), which implies that the model outputs between layers may be similar. This suggests that we can save inference time by sharing parameters between layers to reduce communication overhead. A related algorithm, the "immediate block-wise weight sharing" strategy proposed recently by MobileLLM (Liu et al., 2024), also supports this idea. They share the weights between two adjacent layers to avoid frequent parameter movement in memory (Figure 1(b)) and then train a new small language model. Note that for mobile devices, the communication overhead in memory accounts for a major proportion of the latency overhead, therefore, they double the depth of the new model and obtain better downstream performance, but only increase a negligible additional inference time. However, although MobileLLM achieves significant improvements in accelerating model inference on mobile devices, they focus only on training a new model from scratch and do not fit our purpose of deploying pretrained models through a more resource-saving post-training process.

**Our Contributions.** To apply the layer-sharing strategy to existing LLMs, in this paper, we propose a new layer-sharing algorithm named *SHARP (SHaring Adjacent layers with Recovery Parameters)*, which uses additional low-rank weights to predict subsequent layers, and thus save the memory load overhead of the predicted layers. We summarize our contributions as follows.

- First, we show that language models are robust to the replacement of adjacent or even later MLP layers, which further supports the insight of the layer-sharing strategy. Further, we find that the current layer can be a good approximation of later layers if we add some additional LoRA (Hu et al., 2021) parameters and fine-tune them on a high-quality dataset (Figure 2). We also note that although the outputs of the adjacent layers are similar, their parameters differ a lot. (Section 2.1).
- Second, based on these observations, we propose our SHARP algorithm (Figure 1 (d)). Given the current layer, we use candidate transformations, such as LoRA addition, to predict the parameters of the next several layers using recovery parameters, which reduces memory load overhead and

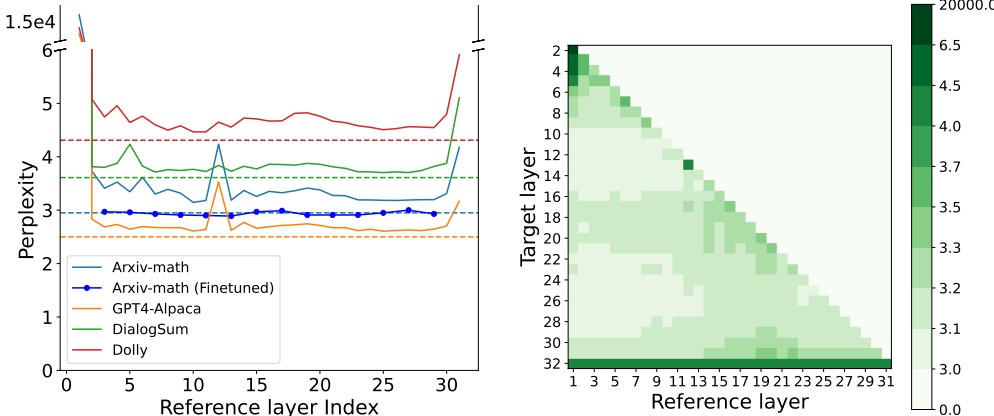

Figure 2: Language models are robust to the replacement of adjacent MLP layers. (Left) For each reference layer, we directly replace the MLP layer in the subsequent layer with that of the reference layer, then evaluate perplexity on various tasks. We find that, aside from the first and last layers, most replacements do not significantly increase perplexity compared to the original model (dotted line). If we fine-tune the model with additional low-rank learnable parameters (rank = 400) added to the next layer, the perplexity gap is effectively closed (as shown by the "Arxiv-math (Finetuned)" line). (Right) Similarly, we observe consistent perplexity results on Arxiv-math (baseline perplexity = 3.0) when using more general reference-target replacement pairs (i.e., use reference layer to replace any later layer).

thus accelerates inference. SHARP consists of two stages: the Single Layer Warmup (SLW) stage and the Supervised Fine-Tuning (SFT) stage, which are used to tune the additional parameters. During SLW, we minimize the $\mathcal{L}_2$ loss between the output of the original replaced layer and the predicted layers, providing a good initialization for the SFT stage. While SFT is critical for recovering model performance, SLW plays an essential role in allowing one layer to aggressively predict multiple layers. With SLW, we can recover model perplexity effectively, even when dropping 3/4 of the original MLP layers (Section 2.2.2, 3.2).

- Third, we conduct detailed ablation studies on SHARP. Specifically, we investigate how to achieve more efficient recovery by determining which layers to replace and selecting the best candidate functions. We find that the earlier layers (layers 5 to 15) are crucial for model performance, and thus, to preserve the model's capacity, more layers should be replaced toward the latter parts of the model. Additionally, we explore various candidate functions, including the vanilla LoRA addition function and other parameterizations, such as left, right, and dot multiplication of the original weights. Interestingly, we find that these different parameterizations perform similarly when their total number of parameters is the same (Sections 2.2.3, 3.3).

- Lastly, we perform several experiments to demonstrate the advantages of our SHARP algorithm. In in-distribution tasks, we recover model perplexity across various tasks, such as Arxiv-math (Kenny, 2023) and Dialogsum Chen et al. (2021), using no more than 50k fine-tuning examples, while saving 38%-65% of the MLP layer parameters. We also show that SHARP performs better on memorization-related downstream tasks, and we discuss how knowledge from different types of downstream tasks is stored in different layers. More importantly, evaluation results on mobile devices demonstrate that SHARP saves **42.8%** in model storage (and loading) and reduces total run time by **42.2%** compared to the original Llama2-7b (Section 3.2, 3.4, 3.5)

## 2 MAIN METHODS

### 2.1 INSIGHT: THE CURRENT LAYER CAN BE A GOOD APPROXIMATION OF THE NEXT LAYER WITH RECOVERY PARAMETERS

As illustrated in the introduction parts, the adjacent layer-sharing strategy (Figure 1 (b)) proposed by MobileLLM accelerates the inference of a newly-trained small language model (125-350M) by reusing the previous layers and reducing the communication overhead in memory. However, several questions about layer-sharing strategies have yet to be revealed: does the success of this adjacent layer-sharing strategy come from the fact that adjacent layers have similar parameters or behaviors?

Can we extend this method to a pretrained larger model like Llama2-7b (Touvron et al., 2023), to accelerate model inference (as shown in Figure 1 (a) and (c))? Further, can we even use one layer to predict more layers and thus further accelerate inference?

To answer these questions, we try to directly replace the MLP layer in the next layer with the current reference layer in Llama2-7b model, and then evaluate its perplexity on some evaluation tasks including Arxiv-math (Kenny, 2023), GPT4-Alpaca (Peng et al., 2023), Databricks-Dolly-15k (Conover et al., 2023) and Dialogsum (Chen et al., 2021). The result is shown in Figure 2 (Left). We found that except for the first and last layer, most replacements don't increase the model perplexity significantly (solid line) compared to the original model (dash line).

More surprisingly, for Arxiv-math, we find that if we add some additional learnable parameters[1] to the replaced layer like Figure 1 (d) and use 50k instruction data from Arxiv-math to finetune them, we can recover the perplexity gap (as the "Arxiv-math (recovered)" line). Note that the Llama2 models use 2T tokens for pretraining, we claim that these results support *it's possible to apply* SHARP *in a larger language model without pretraining-level resources.* Besides in Figure 2 (Right), we also try more general reference-target replacement pairs, i.e., use one reference layer to predict the later layers, and we obtain similar results on the small perplexity gap. This implies that *it's possible to use one reference layer to predict more layers*.

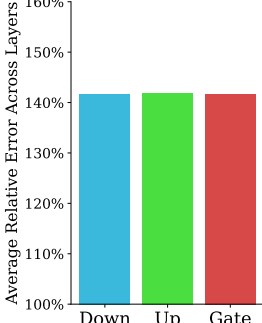

Figure 3: Average relative error between adjacent layers (mean of $\{\|\Theta_{i+1} - \Theta_i\|/\|\Theta_i\|\}_{i=1}^{31}$) are all about 142% for gate, up and down projections.

What's more, it's also worth noting that even though layers with recovery parameters can be good approximations of each other, the parameters themselves are quite different. This is supported by evaluating the average relative error between adjacent layers as shown in Figure 3. This phenomenon implies that to predict the next layer, we should find weights in parameter space to approximate the outputs of adjacent layers, rather than directly approximate their differing parameters.

## 2.2 OUR METHODS

In this section, we illustrate how we find the proper additional parameters for predicting each layer in SHARP. First, we introduce the notations and the settings used in our paper.

### 2.2.1 PRELIMINARY

We assume the original parameters of a particular function (for example, the gate projection of MLP layers) at $i$-th layer as $\Theta_i \in \mathbb{R}^{d_1 \times d_2}$ ($i \in [N]$), where $d_1$ and $d_2$ are the input/output dimension of the function, and we freeze their gradients during training. We denote the low-rank learnable LoRA parameters at $i$-th layer as $A_i \in \mathbb{R}^{d_1 \times r}, B_i \in \mathbb{R}^{r \times d_2}$. And we let $f(\cdot\,;\Theta_i)$ denote the particular function that utilizes $\Theta_i$ as parameters. In this paper, we focus on reducing the parameters in MLP layers, which take up the main parts of the parameters in the intermediate layers. And we use Llama2-7b (Touvron et al., 2023) as our basic model.

To accurately illustrate the replacement, we denote $\mathcal{J} := \{j_1, \ldots, j_K\} \subset [N]$ as the *reference layer sets*[2]. And for each $j_k \in \mathcal{J}$, we define a corresponding continuous sequence named *target layer sets* $\mathcal{T}_{j_k} = [j_k + 1, \ldots, j'_k]$ where $j'_k \in [j_k + 1, j_{k+1} - 1]$. We also denote $g$ to be the candidate transformation function (as in Figure 1 (d)). Our goal is to use each reference layer $j \in \mathcal{J}$ to approximate every later target layer $l \in \mathcal{T}_j$ with some learnable low-rank parameters, i.e., aiming to find some $\Delta\Theta_l$, such that $f(\cdot\,;g(\Theta_j, \Delta\Theta_l))$ performs like $f(\cdot\,;\Theta_l)$. We note that only the reference layer will be loaded and stored, and target layers will be obtained using the corresponding reference layer combining low-rank additional parameters on the fly. In this way, we trade cheaper parameter computation for more expensive memory loading.

---

[1]Here $g$ is the LoRA addition, i.e., $g(\Theta, (\alpha, U, V)) = \alpha\Theta + U \cdot V$, where $\alpha \in \mathbb{R}, \Theta \in \mathbb{R}^{d_1 \times d_2}, U \in \mathbb{R}^{d_1 \times r}, V \in \mathbb{R}^{r \times d_2}$. And here we use rank=400, which is much less than 4096, the rank of original parameters.

[2]Since from definition, each reference layer will be reused in the later layer but not another reference layer, obviously we have $j_{l+1} - j_l >= 2, \forall l \in K - 1$.

**Stored Ratio $\tau$.** We define $\tau$ to denote the ratio of the stored layers in the replaced model to that in the original model. For instance, if we consider Llama2-7b, which has 32 layers, and the number of replaced layers is $X$, then $\tau := (32 - X)/32$.

**Compression Ratio $s$.** We define $s$ to represent the ratio of the parameters of the MLP layer in the replaced model to that in the original model. For instance, assume that we use the LoRA addition function as candidate transformation, the rank of additional weight is $r$, and the number of replaced layers is $X$. Then if we consider Llama2-7b, whose MLP weights have the dimension $4096 \times 11008$, then the compression ratio can be calculated by

$$s = \frac{32 - X}{32} + \frac{X}{32} \times \frac{4096r + 11008r}{4096 \times 11008} \approx 1 - \frac{X}{32} + X \cdot r \times 10^{-5} \tag{1}$$

### 2.2.2 SHARP ALGORITHM

In this part, we show how we achieve SHARP (Figure 1 (d)) algorithm. First, we illustrate why we choose such a two-stage algorithm for recovering.

**Why Two-Stage?** A natural way to recover the model performance is to directly finetune the model end-to-end for all learnable low-rank additional parameters. In general, it works fine when we try to use one layer to replace the next layer, which at most gives a 50% reduction of the MLP layers. However, if we want to use one layer to compute multiple adjacent layers, just using the SFT stage will require more data for recovering, and result in a much slower convergence rate and worse final result. The detailed discussion of this has been investigated in Section 3.3.3 and Table 3, and the intuition for this phenomenon may be that when we use one layer to replace multiple layers aggressively, the model loses too many parameters and thus start optimizing from an initialization point that is far from the optimal solution. Therefore, we need first to align the output of the predicted and original target layers before the SFT stage. Details of the algorithm are as follows.

**Stage 1: Single Layer Warmup (SLW).** First, we minimize the $\mathcal{L}_2$ loss between the output of layer predicted by the reference layer and that of the original target layer by finetuning the model on high-quality data. Formally, for each reference layer $j \in \mathcal{J}$ and every target layer predicted by this reference layer $l \in \mathcal{T}_j$, we want to find:

$$\Delta\Theta_l^1 \leftarrow \arg\min_{\Delta\Theta_l} \mathbb{E}_{X \sim \mathcal{P}^l} \left[ \|f(X; g(\Theta_i, \Delta\Theta_l)) - f(X; \Theta_l)\|_2^2 \right] \tag{2}$$

Here $\mathcal{P}_l$ is the distribution of the input activations of $f(\cdot\,; \Theta_l)$, and it can be obtained by running the forwarding pass of the original model on the finetuning dataset. We also note that the SLW stage of each target layer can be much faster than the SFT stage (Stage 2) since we just run one MLP layer, which has only 135M parameters (while the whole model has 7B parameters). And this process can also be fully parallelized since the SLW stages of different target layers are independent. In Section 3.3.3, we will show that SLW is critical for increasing the compression ratios.

**Stage 2: Supervised Fine-Tuning (SFT).** After the single MLP warmup stage, we partly recover the output of the replaced layers. To better align different replaced layers together and obtain better model output, at the second stage, we fixed the original parameters and finetune all the learnable low-rank components $\{\Delta\Theta_*\}$ together. In Section 3.3.3, we will also show that although SLW is important, SFT is still the key stage to recover the model capacity.

### 2.2.3 CHOICE OF REPLACEMENT AND CANDIDATE TRANSFORMATION

**Replacement Type.** Note that in SHARP, there are multiple ways to define the reference layer set $\mathcal{J}$ and the corresponding target layer set $\mathcal{T}$. To prevent ambiguity, we formally list the types of layer replacement that we used in this paper in Table 1. And more detailed table are in Table 8.

Notably, we skip the first and the last layer for all types, since Figure 2 shows that these two layers may be quite important or behave differently from other layers. For $T_{next}$ and $T_{next2}$, we consider the constant reference intervals (each reference predicts the next or next two target layers). For $T_{back}$ and $T_{front}$, we consider the cases where making one reference layer predict more target layers in the front parts and back parts, respectively. And finally, for $T_{more}$ and $T_{max}$, we aggressively try to remove more MLP layers. In Section 3.3.1, we will show that repeating layers in the back parts of the model is better than doing this in the front parts, i.e., $T_{back}$ is better than $T_{front}$, and even better than $T_{next2}$. Furthermore, we will show that aggressively removing the layers like $T_{more}$ and $T_{max}$ can still achieve quite good recovery performance.

Table 1: Different replacement types. Here the stored ratio $\tau$ is defined in Section 2.2.1. For example in $T_{next}$, we need to store 18 layers (1,2,3,5,...29, 31,32), out of 32 layers in the original model, so $\tau = 56\%$. More details about the reference layers and the target layers are shown in Table 8.

| Type | Stored Ratio $\tau$ | Description |
|---|---|---|
| $T_{next}$ | 56% | Replacing layer $2t$ with layer $2t-1$ for $t \in [2, 15]$ |
| $T_{next2}$ | 44% | Replacing layer $3t+1, 3t+2$ with layer $3t$ for $t \in [1, 9]$ |
| $T_{back}$ | 38% | Replacing more layers in the *back* parts of the model. |
| $T_{front}$ | 38% | Replacing more layers in the *front* parts of the model. |
| $T_{more}$ | 25% | Replace more aggressively, just store 8 layers |
| $T_{max}$ | 16% | Replace more aggressively, just store 5 layers |

**Candidate Transformation Type.** On the other hand, we also consider different candidate transformations $g$. In detail, we investigate the following parameterization ways:

$$g_0(\Theta_j, (\alpha, A_l, B_l)) := \alpha\Theta_j + A_l B_l, \quad \alpha \in \mathbb{R}, A_l \in \mathbb{R}^{d_1 \times r}, B_l \in \mathbb{R}^{r \times d_2} \tag{3}$$

$$g_1(\Theta_j, (\alpha, A_l, B_l, C_l, D_l)) := \alpha\Theta_j C_l^\top D_l + A_l B_l, \quad \alpha \in \mathbb{R}, A_l, E_l, F_l \in \mathbb{R}^{d_1 \times r}, B_l, C_l, D_l \in \mathbb{R}^{r \times d_2} \tag{4}$$

$$g_2(\Theta_j, (\alpha, A_l, B_l, E_l, F_l)) := \alpha E_l F_l^\top \Theta_j + A_l B_l, \quad \alpha \in \mathbb{R}, A_l, E_l, F_l \in \mathbb{R}^{d_1 \times r}, B_l \in \mathbb{R}^{r \times d_2} \tag{5}$$

$$g_3(\Theta_j, (\alpha, A_l, B_l, U_l, V_l)) := \alpha[(U_l V_l) \odot \Theta_j] + A_l B_l, \quad \alpha \in \mathbb{R}, A_l, U_l \in \mathbb{R}^{d_1 \times r}, B_l, V_l \in \mathbb{R}^{r \times d_2} \tag{6}$$

Here we consider the vanilla LoRA, left multiplication, right multiplication, and dot multiplication. We assume $d_1 = 4096 < d_2 = 11008$ for Llama2-7b to prevent ambiguity. The comparison result is shown in Section 3.3.2. Surprisingly we find these four different transformations have almost the same capability for recovering model performance if their numbers of parameters are the same.

## 3 EXPERIMENTS

In the experiment section, we mainly focus on four parts: **(E1)** In-distribution recovery tasks, where we apply SHARP to the pretrained model, finetune it on a specific high-quality dataset, and then evaluate the model's perplexity on the same dataset. **(E2)** Ablation study, where we investigate how to choose better replacement types and candidate transformations. We also try to analyze how the single-layer warmup stage and different settings influence the model recovery, and want to see how aggressively we can choose the replacement type. **(E3)** Downstream evaluation, where we assess the model's performance on several widely-used downstream tasks. **(E4)** Latency analysis, where we examine how much inference acceleration SHARP can achieve.

### 3.1 EXPERIMENTAL SETTINGS

**Dataset.** We use several high-quality datasets in our in-distribution recovery tasks: (1) Arxiv-math (Kenny, 2023): it includes 50k high-quality QA instruction data from the mathematical domain. (2) GPT4-Alpaca (Peng et al., 2023): it contains 52k English instruction-following generated by GPT-4 using Alpaca prompts. (3) Databricks-Dolly (Conover et al., 2023): This is an open-source dataset comprising 15k instruction-following records generated by Databricks employees. (4) DialogSum (Chen et al., 2021): this is a dialogue summarization dataset consisting of 13.5k dialogues (12.5k for training) with corresponding manually labeled summaries and topics. (5) OpenOrca (Mukherjee et al., 2023), which is a collection of augmented FLAN Collection data (Longpre et al., 2023), containing about 1M GPT-4 completions and about 3.2M GPT-3.5 completions. We select a 50k subset of it for in-distribution tasks.

In downstream evaluation parts, we also use (6) FineWeb-Edu (Penedo et al., 2024; Lozhkov et al., 2024), consists of 1.3T tokens of educational web pages filtered from the FineWeb dataset. We use its "sample-10BT" subset. (7) Tulu V2 Collection (Ivison et al., 2023), which contains 326k instruction data mixed from FLAN (Longpre et al., 2023), Open Assistant 1 (Köpf et al., 2024), GPT4-Alpaca (Peng et al., 2023), Code-Alpaca (Chaudhary, 2023), LIMA (Zhou et al., 2024), WizardLM (Xu et al., 2024) and Open-Orca (Mukherjee et al., 2023).

**Finetuning Model.** Here we use Llama2-7b (Touvron et al., 2023) as the basic model. For the Single Layer Warmup (Stage 1), we use the Adam optimizer with a fixed learning rate of 1e-3

Table 2: The perplexity of different methods and different replacement types in the In-distribution recovering tasks. The smaller value the better. We use rank=400 for the additional parameters and let $g_0$ (Eqn 3) be the candidate transformation. The compression ratio $s$ is defined in Section 2.2.1. "SHARP (w/o fine-tuning)" means just use the first Single Layer Warmup Stage for recovering.

| Models | Type | $s$ | Arxiv-math | DialogSum | GPT4-Alpaca | Dolly | OpenOrca |
|---|---|---|---|---|---|---|---|
| Original | | 100% | 3.0 | 3.7 | 2.5 | 3.6 | 4.5 |
| Direct Sharing | $T_{next}$ | 62% | 2171.3 | 801.7 | 20662.1 | 7221.7 | 12108.5 |
| SHARP (w/o fine-tuning) | $T_{next}$ | 62% | 4.8 | 5.3 | 4.2 | 7.2 | 8.4 |
| SHARP | $T_{next}$ | 62% | **3.2** | **3.8** | **2.8** | **4.7** | **4.3** |
| Direct Sharing | $T_{back}$ | 46% | 92603.9 | 154262.4 | 67908.4 | 136787.2 | 86993.3 |
| SHARP (w/o fine-tuning) | $T_{back}$ | 46% | 7.7 | 7.6 | 7.9 | 13.7 | 13.5 |
| SHARP | $T_{back}$ | 46% | 3.5 | 4.2 | 3.2 | 5.8 | 4.9 |
| Direct Sharing | $T_{more}$ | 35% | 318430.3 | 143335.7 | 346485.1 | 280953.7 | 300136.8 |
| SHARP (w/o fine-tuning) | $T_{more}$ | 35% | 26.6 | 9.8 | 11.3 | 23.7 | 21.8 |
| SHARP | $T_{more}$ | 35% | 3.7 | 4.4 | 3.4 | 6.4 | 5.3 |

for 5 epochs. For the SFT (Stage 2), we follow the pipeline of Open-Instruct (Wang et al., 2023) The learning rate of the SFT stage is 2e-5, the warmup ratio[3] of the SFT stage is 5% and the max sequence length is 2048. In the in-distribution recovering tasks, we random sample 10% of the training data (except 30% for Databricks-Dolly and 5% for OpenOrca) to calculate the activations in the intermediate layers for the SLW (Stage 1). While in the downstream evaluation tasks, we use 10% of the Arxiv-math data for Stage 1, and then use 50k instruction data from Arxiv-math, 200k data from FineWeb-Edu, and all data that is from Tulu V2 and listed above for SFT (Stage 2). The total number of tokens used for finetuning at the downstream evaluation tasks is about 0.7B, which is much less than that of the pretraining data used by Llama2 (2T).

**Evaluation.** For the in-distribution recovering tasks, we select 1% of the data from each task for calculating perplexity, while the other 99% for recovering model performance. For the downstream evaluation, we follow the pipeline of Language Model Evaluation Harness (Gao et al., 2024). We select the evaluation tasks that are widely used in other works and measure various abilities of the model, including memorization and reasoning. Details are illustrated in Appendix C.1.

### 3.2 E1: In-Distribution Recovering tasks

In this section, we use rank=400 for the additional parameters employed in SHARP. It is important to note that the projection matrices in the MLP layers of Llama2-7b have a shape of $4096 \times 11008$, meaning the additional parameters occupy no more than 13% of the original parameter size.

We show the results of in-distribution recovering tasks in Table 2. Directly sharing adjacent layers, as in Figure 1 (c) (Liu et al., 2024), leads to unacceptably high perplexity and meaningless output. However, using the Single Layer Warmup (SHARP w/o fine-tuning), the perplexity of the model gets to a reasonable range. After applying SFT with the in-distribution data, the perplexity gap between the original and the SHARP-processed model can be further reduced. In particular, $T_{next}$ almost fixes the gap for most of the tasks except Databricks-Dolly. What's more, after radically giving up more than 3/4 of the layers ($T_{more}$, compression ratios are calculated by Eqn. 1), we can still recover the perplexity gap quite well. These results show the potential of SHARP to save model parameters and accelerate inference.

### 3.3 E2: Further ablation study

In this section, we conduct several ablation studies of SHARP.

#### 3.3.1 How to Replace Better?

---

[3]Note this is the warmup steps of SFT rather than the SLW stage

Table 3: Different settings for SHARP. It shows the model perplexity evaluated on Arxiv-math, where the baseline result is 3.0. We use $g_0$ (Eqn. 3) as the candidate function, and the Types are shown in Table 1 The percentage in the bracket means the proportion of the Arxiv-math training data that are randomly sampled for the SFT stage. Compression ratios $s$ is calculated by Eqn. 1.

| Rank $r$ | $r = 5$ | $r = 20$ | $r = 400$ | $r = 5$ | $r = 20$ | $r = 400$ | $r = 400$ |
|---|---|---|---|---|---|---|---|
| **Type** | | $\mathbf{T_{next}}$ | | | $\mathbf{T_{back}}$ | | $\mathbf{T_{more}}$ |
| Compression Ratio $s$ | 56% | 57% | 62% | 38% | 38% | 46% | 35% |
| Direct Sharing | 2171.3 | 2171.3 | 2171.3 | 92603.9 | 92603.9 | 92603.9 | 318430.3 |
| SHARP (w/o fine-tuning) | 13.0 | 9.5 | 4.8 | 73.7 | 33.9 | 7.7 | 26.6 |
| Supervised Fine-Tuning (10%) | 4.9 | 5.0 | 5.0 | 6.2 | 6.7 | 8.6 | 14.8 |
| Supervised Fine-Tuning (100%) | **4.0** | _4.0_ | _3.8_ | **4.5** | **4.2** | _3.6_ | 6.8 |
| SHARP (10%) | 4.9 | 4.9 | 3.9 | 6.0 | 5.6 | 4.2 | _4.5_ |
| SHARP (100%) | _4.1_ | **3.8** | **3.2** | _4.8_ | **4.2** | **3.5** | **3.7** |

First, we investigate that with the same or similar number of the stored layers (i.e., stored ratio $s$), which type of replacement can better retain model capacity. Like the previous section, we use matrices with rank=400 as the additional LoRA parameters, and we try all the SHARP types shown in Table 1. The results are in Table 4. We find that interestingly, although $T_{next}$ recovers the perplexity quite well, $T_{next2}$ only has the same or even worse results than $T_{back}$, which has a smaller stored ratio ($s$). Similarly, $T_{front}$ has worse performance than $T_{back}$. These results show that *replacing more layers at the back parts of the model can better maintain the model performance*. The discussion of Figure 4 in Section 3.4 will further support this claim. Furthermore, we can see that $T_{more}$ and $T_{max}$ also have impressive recovery results, even though they drop most of the MLP layers, which also support our claims in Section 2.1.

Table 4: Recovery results for different SHARP Types. Store ratio $\tau$ is defined in Section 2.2.1

| Type | $\tau$ | Arxiv-math | DialogSum |
|---|---|---|---|
| Baseline | 100% | 3.0 | 3.7 |
| $\mathbf{T_{next}}$ | 56% | 3.2 | 3.8 |
| $\mathbf{T_{next2}}$ | 44% | 3.7 | 4.2 |
| $\mathbf{T_{back}}$ | 38% | 3.5 | 4.2 |
| $\mathbf{T_{front}}$ | 38% | 3.8 | 4.4 |
| $\mathbf{T_{more}}$ | 25% | 3.7 | 4.4 |
| $\mathbf{T_{max}}$ | 16% | 4.1 | 4.9 |

### 3.3.2 DIFFERENT CANDIDATE TRANSFORMATION

In this part, we compare various candidate transformations listed in Section 2.2.3, i.e., Equation (3),(4),(5), and (6), to assess their impact on model recovery. For a fair comparison, we adjust ranks so all candidates have almost the same number of additional parameters. The setting and the result are shown in Table 5. Surprisingly, except for the SLW stage of $g_1$, all other candidate transformations have similar performance for both SLW and the complete SHARP stages. This may show that using the same amount of additional parameters, different parameterization strategies have the same

Table 5: Perplexity on Arxiv-math for different candidate transformations ($T_{next}$).

| Type | rank | SHARP (w/o fine-tuning) | SHARP |
|---|---|---|---|
| Baseline | NA | NA | 3.0 |
| $g_0$ (Eqn. 3) | 400 | 4.8 | **3.2** |
| $g_1$ (Eqn. 4) | 163 | 5.5 | 3.3 |
| $g_2$ (Eqn. 5) | 259 | 4.8 | 3.3 |
| $g_3$ (Eqn. 6) | 200 | **4.7** | 3.3 |

recovery capabilities. And since LoRA-style transformation, $g_0$ has the simplest forms and more comprehensive studies, we choose $g_0$ as our candidate transformation in the later experiment parts.

### 3.3.3 ABLATION STUDY FOR DATASET SIZE, RANK AND TYPE

To find the optimal algorithm settings, we further explore how different choices of dataset size, rank of the additional parameters, and the SHARP Types together influence the recovery performance. We choose Arxiv-math as the in-distribution task and select ranks among 5, 20, and 400. The methods contain the vanilla adjacent layer sharing (Direct Sharing), just use the first stage of SHARP (SLW, SHARP w/o fine-tuning), just use the second stage (SFT), and the complete SHARP. Training dataset size includes all the dataset (100%) or a small random subset of it (10%).

Table 6: Recover results on the downstream tasks. The higher the better for all the tasks. Here we use rank=400 and the replacement type is $T_{next}$, therefore, the compression ratio is 62%.

| Models | SciQ | BoolQ | PIQA | ARC-E | ARC-C | WinoGrande | MedMCQA |
|---|---|---|---|---|---|---|---|
| Original | 94.1 | 77.8 | 78.1 | 76.3 | 43.3 | 69.4 | 34.5 |
| Direct Sharing | 22.7 | 49.1 | 56.7 | 26.8 | 22.0 | 50.6 | 22.4 |
| SHARP (w/o fine-tuning) | 87.8 | 66.1 | 63.0 | 54.8 | 26.8 | 57.8 | 30.5 |
| SHARP | 92.6 | 76.0 | 72.6 | 65.8 | 34.7 | 62.4 | 31.4 |

| | BBH | GSM8k | MATHQA | MUTUAL | LAMBADA | ComSenQA | QA4MRE |
|---|---|---|---|---|---|---|---|
| Original | 38.6 | 14.2 | 28.4 | 70.8 | 71.9 | 32.6 | 42.8 |
| Direct Sharing | 0.0 | 0.0 | 20.7 | 57.5 | 8.1 | 20.0 | 16.6 |
| SHARP (w/o fine-tuning) | 24.8 | 1.6 | 22.1 | 59.8 | 35.0 | 20.0 | 30.2 |
| SHARP | 29.8 | 3.6 | 24.7 | 68.3 | 58.3 | **44.1** | 38.3 |

The results are shown in Table 3. We note that although the performance of SLW alone is not advanced, it is beneficial for the whole SHARP process. *Especially, the larger the rank, the smaller the finetuning dataset size, and the more aggressively we drop the layers, the more important the SLW stage serves.* For example, in (r=400, $T_{back}$) case, SFT (10%) just recovers the perplexity to 8.6, while SLW + SFT (10%) can improve it to 4.2. And in (r=400, $T_{more}$) case, even SFT (100%) performs much worse than SLW + SFT (100%). In general, SLW + SFT (10%) already approaches the best result SLW + SFT (100%) quite well although they just use 10% of the finetuning data. On the other hand, when the rank is small, the influence of SLW is not significant. As in rank=5, SLW + SFT has similar results to SFT alone for both $T_{next}$ and $T_{back}$. We think the reason for this is that the target of the SLW stage is to provide a better LoRA weight initialization for the SFT. Since we use $\mathcal{L}_2$ loss at SLW, the warmup performance becomes better when the rank of additional parameters becomes larger (Like for $T_{back}$, SLW alone with r=5 just improves perplexity to 73.7, while r=400 can bring it to 7.7), and its gain becomes more obvious when more layers need to be predicted and the SFT data is not enough.

Besides, we also find that in general, SFT is the key stage of SHARP to recover the final performance, and a larger rank and larger finetuning dataset brings better performance. For the best setting (rank=400, SLW + SFT (100%)), aggressively dropping most of the MLP layers like $T_{more}$, which drops 75% of the original MLP layers, still keeps quite good performance (3.7) in perplexity compared to the best result of $T_{next}$ (3.2).

### 3.4 E3: DOWNSTREAM EVALUATION

Here we use type $T_{next}$ with rank=400 for recovering parameters. Due to time and resource constraints, we fine-tuned the model using only 0.7B tokens, significantly fewer than Llama2's 2T pre-training tokens and the 30B–150B tokens used in sparsification works (Mirzadeh et al., 2023; Song et al., 2024). However, it still brings positive signals and further understanding of our methods. The result is shown in Table 6.

As expected, SHARP greatly outperforms Direct Sharing and SLW alone on all the evaluation tasks. In particular, we notice that SHARP performs relatively better in tasks requiring knowledge memorization capability. For example, compared to the original model, SHARP has very close evaluation results on SciQ and BoolQ, which focus on memorizing scientific concepts and knowledge from various domains, respectively. Notably, it even surpasses the baseline on CommonsenseQA (44.1 versus 32.6), which measures the commonsense knowledge of the model. On the other hand, for tasks that further require more complex reasoning capabilities, like GSM8k, ARC-Easy, ARC-Challenge, and PIQA, the performance gap between SHARP and the original model is still large.

For a better understanding of why SHARP has different performance on different kinds of evaluation tasks, we try to observe the sensitivity of different layers to each task. To achieve this, we set the parameters of each MLP layer to zero and compare the evaluation results between the original model and the modified model. The results are shown in Figure 4, and we have two observations: (1) We find that layers 5 to 15 and the last two layers are important for all tasks. Setting their parameters to 0 will hurt both knowledge-memorization tasks and reasoning tasks. This supports our claim in Section 3.3.1 that keeping the front parts of the layers can retain more model capacity.

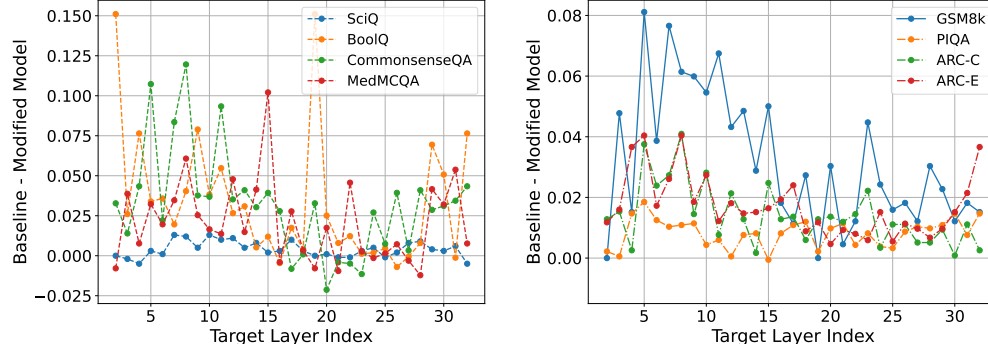

Figure 4: Impact of different layers on model capabilities. The x-axis denotes the index of the zero-out MLP layer, whose weights are set to be zero, in the modified model, and the y-axis shows the difference between the original model and the modified model on the particular evaluation tasks, which means that the lower the value the better. (Left) Evaluation tasks focused on memorizing domain-specific knowledge or common sense. (Right) Evaluation tasks requiring reasoning abilities in areas like mathematics, physics, or general reasoning. We skip index 0 since it's critical based on Figure 2.

(2) Interestingly, for layers 16 to 30, we find different phenomenons for knowledge memorization tasks and reasoning tasks. For knowledge memorization tasks, just a few layers between 16 to 30 are important for the performance, and they may vary for different tasks. For example, 19 and 29 for BoolQ, 19,24,26, and 27 for CommonsenseQA, and 22 and 29 for MedMCQA. While for the complex reasoning tasks, almost all the layers between 16 and 30 have a non-negligible influence on the model output. These may imply that the model uses some specific weights in the later layers to memorize knowledge from various domains, while for more complex reasoning tasks, the capability is related to all the layers, which results in it needing more data and resources for the model to recover performance. This ablation study will also be useful for future works of interpreting how models capture knowledge and capabilities at each layer.

Table 7: Run time results (seconds) on mobile.

| Model | Load & Init (std) | Forward (std) | Total time | Model size |
|---|---|---|---|---|
| Original Llama2-7b | 9.794 (0.627%) | 2.905 (1.573%) | 12.699 | 4.04GB |
| SHARP ($T_{next}$) | 5.684 (0.645%) | 1.630 (1.363%) | 7.314 | 2.31GB |
| SHARP ($T_{next}$) saving | 42.0% | 43.9% | 42.2% | 42.8% |

## 3.5 E4: LATENCY ANALYSIS

In this section, we measure the time to load and initialize the model and the average time to run a single forward over 5 runs using ExecuTorch v0.3.0 on iPhone 16 Pro (iOS 18.1), with XNNPack backend. To enable to model to be fitted into the phone with 8GB RAM, we use 4-bit integer quantized models. Detailed experimental setting is in Appendix C.2. Results in Table 7 reflect that through weight sharing, SHARP saves 42.0% of loading and initialization time, which is the dominant of the model execution, attributable to fewer layers required to store and load. SHARP also runs faster than the original Llama2-7b by 43.9%, benefiting from data locality. Overall, our SHARP saves 42.2% run time and 42.8% model storage compared to the original Llama2-7b.

## 4 CONCLUSION

This paper presented SHARP, a method to accelerate large language model inference by sharing adjacent layers with recovery parameters. SHARP effectively reduces model size and inference time by using the reference layer to predict the later layers, which saves the memory load overhead. It recovers the model performance through a two-stage process: Single Layer Warmup and Supervised Fine-Tuning. Experimental results demonstrate that SHARP achieves comparable perplexity to original models across various in-distribution tasks. By minimizing the number of layers and parameters needed, SHARP provides a practical solution for deploying large models in resource-constrained environments. We believe this method can further enlighten researchers in designing advanced layer-sharing methods for accelerating inference and may be insightful for the interpretability-related works on understanding how each layer works in LLM.

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

## A  RELATED WORK

**Model compression**  Model compression is a classical way to improve inference efficiency by either reducing the number of model parameters or lowering the memory required to store them. Three widely used techniques—sparsity and pruning, quantization, and distillation—are central to this goal. Sparsity and pruning share the aim of reducing the number of effective parameters, but differ in approach: sparsity reduces individual weights to zero in an unstructured manner (Sun et al., 2023; Xia et al., 2023a; Frantar & Alistarh, 2023), while pruning takes a structured approach by removing entire components, such as neurons or filters, from the network (Xia et al., 2023b). Quantization reduces the memory footprint by lowering the precision of weights and activations, without changing the number of parameters (Dettmers et al., 2024; 2022; Li et al., 2023a; Kim et al., 2023a; Frantar et al., 2022; Xiao et al., 2023; Yao et al., 2022; Liu et al., 2023a; Frantar et al., 2022; Zhang et al., 2018). While sparsity, pruning, and quantization are usually applied after a certain amount of training, distillation is a data-centric methodology used during training. In distillation, a smaller student model is trained using both the original training data and the output (soft labels) of a larger teacher model, allowing the student model to retain much of the teacher's performance while being more efficient (Hinton, 2015; Timiryasov & Tastet, 2023; Chen et al., 2024). These three categories represent the most classical methods for model compression, primarily aimed at improving inference efficiency. However, there are other compression methods closely related to our proposed techniques, which will be discussed in detail later.

**Low rank approximation**  Low-rank approximation, while distinct from the traditional model compression techniques discussed earlier, leverages the observation that much of the key information users care about in a neural network can be represented in a lower-dimensional subspace. By approximating large weight matrices with low-rank representations, both the number of parameters and computational costs are reduced. Many works such as Li et al. (2023b); Hsu et al. (2022); Hajimolahoseini et al. (2021); Tahaei et al. (2021) focus on improving inference efficiency using this method, but it can also offer significant efficiency gains during training. LoRA (Low-Rank Adaptation) by Hu et al. (2021) is the first work to introduce two small low-rank matrices, $A$ and $B$ attached to a frozen pre-trained weight matrix $W$, allowing for efficient fine-tuning with minimal memory usage. Since then, numerous variants have been developed to enhance this approach (Dettmers et al., 2022; Sheng et al., 2023; Chen et al., 2023; Zhang et al., 2023). Our proposed methods are similarly inspired by low-rank approximation, but unlike other works that focus on decomposing the entire weight matrix, we use low-rank approximations to estimate the minimal differences between intermediate layers. This allows for maximal weight sharing, significantly reducing redundancy while maintaining performance.

**Weight sharing**  Weight sharing is another powerful model compression technique that improves both training and inference efficiency by reducing the number of unique parameters in a neural network. Classical weight sharing involves using a common representation space across multiple tasks or domains, allowing models to generalize better while using fewer parameters (Liu et al., 2020; Jiang et al., 2019; Tars & Fishel, 2018; Fu et al., 2021). Those embedding sharing architectures have been later adopted in Llama (Touvron et al., 2023) and OPT models (Zhang et al., 2022). However the savings from embedding sharing diminished with the increasing model size, and therefore been disregarded in recent designs of LLMs. Recently, MobileLLM(Liu et al., 2024) introduced the first approach to weight sharing between intermediate layers, enabling models to reuse learned representations across layers. This significantly reduces the parameter count while maintaining performance, making large models more feasible for resource-constrained environments. Our proposed methods are inspired by this concept, but further integrate weight sharing with low-rank approximation to achieve both computational efficiency and performance preservation during the fine-tuning stage.

**Small models**  The definition of small models has evolved as advancements in deep learning architectures have significantly increased model sizes. Models that were previously considered large are now categorized as small relative to the current state-of-the-art. Commonly, models with fewer than 7 billion parameters (7B) are referred to as small models. Notably, prominent open-source language models under 7B parameters include Mistral 7B (Jiang et al., 2023); Phi-3 series (Abdin et al., 2024); Gemma 2B (Team et al., 2023), Llama 3.2 series (Dubey et al., 2024), TinyLlama(Zhang et al., 2024a), MobileLLM(Liu et al., 2024) and MiniCPM(Hu et al., 2024). Despite their smaller

Table 8: Different replacement types. $(j : \mathcal{T}_j)$ denotes the reference layer $j$ and the target layers set $\mathcal{T}_j$ which use $j$ to predict their parameters, and we briefly describe how each type replace the layers. Here the stored ratio $\tau$ is defined in Section 2.2.1. For example in $\text{T}_{\text{next}}$, we need to store 18 layers (1,2,3,5,...29, 31,32), out of 32 layers in the original model, so $\tau = 56\%$.

| Type | Stored Ratio $\tau$ | Description and $[(j : \mathcal{T}_j)]$ |
|---|---|---|
| $\text{T}_{\text{next}}$ | 56% | Replacing layer $2t$ with layer $2t - 1$ for $t \in [2, 15]$ |
| | | (**3**:4), (**5**:6), (**7**:8), (**9**:10), (**11**:12), (**13**:14), (**15**:16) (**17**:18), (**19**:20), (**21**:22), (**23**:24), (**25**:26), (**27**:28), (**29**:30) |
| $\text{T}_{\text{next2}}$ | 44% | Replacing layer $3t + 1, 3t + 2$ with layer $3t$ for $t \in [1, 9]$ |
| | | (**3**:4,5), (**6**:7,8), (**9**:10,11), (**12**:13,14), (**15**:16,17), (**18**:19,20), (**21**:22,23), (**24**:25,26), (**27**:28,29) |
| $\text{T}_{\text{back}}$ | 38% | Replacing more layers in the *back* parts of the model. |
| | | (**3**:4), (**5**:6), (**7**:8), (**9**:10), (**11**:12), (**13**:14,15), (**16**:17,18,19,20,21,22), (**23**:24,25,26,27,28,29,30) |
| $\text{T}_{\text{front}}$ | 38% | Replacing more layers in the *front* parts of the model. |
| | | (**3**:4,5,6,7,8,9,10), (**11**:12,13,14,15,16,17), (**18**:19,20), (**21**:22), (**23**:24), (**25**:26), (**27**:28), (**29**:30) |
| $\text{T}_{\text{more}}$ | 25% | Replace more aggressively, just store 8 layers |
| | | (**3**:4,5,6), (**7**:8,9,10,11), (**13**:14,15,16,17,18,19,20,21,22), (**23**:24,25,26,27,28,29,30) |
| $\text{T}_{\text{max}}$ | 16% | Replace more aggressively, just store 5 layers |
| | | (**2**:3,4,5,6,7,8,9,10), (**11**:12,13,14,15,16,17,18,19,20), (**21**:22,23,24,25,26,27,28,29,30,31) |

size, these models remain challenging to deploy on edge devices due to their computational and memory requirements, necessitating further optimization and compression techniques for deployment in resource-constrained environments.

# B  ALGORITHM DETAILS

## B.1  FULL DEFINITION OF DIFFERENT REPLACEMENT TYPES

Here we show the full table of different replacement types used in our main paper in Table 8.

# C  EXPERIMENT DETAILS

## C.1  EVALUATION TASKS

This section introduces the evaluation tasks used in our downstream evaluation part (Section 3.4).

### C.1.1  BASIC REASONING TASKS

This class of tasks doesn't require too much commonsense or knowledge to solve problems, but needs the model to have good reasoning capability, especially the mathematical reasoning

- **MathQA** (Amini et al., 2019): This is a large-scale dataset of 37k English multiple-choice math word problems covering multiple math domain categories by modeling operation programs corresponding to word problems in the AQuA dataset.

- **GSM8k** (Cobbe et al., 2021): This is a free-generation benchmark of grade school math problems aiming for evaluating multi-step (2-8 steps) mathematical reasoning capabilities. These problems are illustrated by natural language and require using four basic arithmetic operations to reach the final answer.

- **BBH-COT-FS** (Suzgun et al., 2022): This is a free-generation benchmark consists a suite of 23 challenging BIG-Bench tasks (Srivastava et al., 2022) which we call BIG-Bench Hard (BBH). These are the tasks for which prior language model evaluations did not outperform the average human-rater. Here we use the chain-of-though with 3 shot version.

- **MuTual**(Cui et al., 2020): This is a retrieval-based dataset for multi-turn dialogue reasoning, which is modified from Chinese high school English listening comprehension test data.

- **QA4MRE**(Peñas et al., 2013): This is a multi-choice benchmark used for long-text understanding and reasoning. Four different tasks have been organized during these years: Main Task, Processing Modality and Negation for Machine Reading, Machine Reading of Biomedical Texts about Alzheimer's disease, and Entrance Exam.

### C.1.2 KNOWLEDGE-MEMORIZATION TASKS

- **SciQ** (Welbl et al., 2017): This contains 13,679 crowdsourced science exam questions about Physics, Chemistry and Biology, among others. The questions are in multiple-choice format with 4 answer options each. For the majority of the questions, an additional paragraph with supporting evidence for the correct answer is provided.

- **BoolQ** (Clark et al., 2019): This is a question-answering benchmark for yes/no questions containing 15942 examples. These questions are naturally occurring – they are generated in unprompted and unconstrained settings. Each example is a triplet of (question, passage, answer), with the title of the page as optional additional context.

- **CommonsenseQA** (Talmor et al., 2019): This is a multiple-choice question-answering dataset that requires different types of commonsense knowledge to predict the correct answers. It contains 12,102 questions with one correct answer and four distractor answers.

- **MedMCQA** (Pal et al., 2022) This is a new large-scale, multiple-choice question-answering dataset designed to address real-world medical entrance exam questions. More than 194k high-quality AIIMS \ NEET PG entrance exam MCQs covering 2.4k healthcare topics and 21 medical subjects are collected with an average token length of 12.77 and high topical diversity. Each sample contains a question, correct answer(s), and other options which require a deeper language understanding as it tests the 10+ reasoning abilities of a model across a wide range of medical subjects & topics. A detailed explanation of the solution, along with the above information, is provided in this study.

- **TriviaQA** (Joshi et al., 2017): This is a reading comprehension dataset containing over 650K question-answer-evidence triples. TriviaQA includes 95K question-answer pairs authored by trivia enthusiasts and independently gathered evidence documents, six per question on average, that provide high-quality distant supervision for answering the questions. This dataset can be used for both retrieval-augmented models to test the models' knowledge retrieval ability and the usual LLM to test the knowledge memorization on the model itself.

### C.1.3 KNOWLEDGE-MEMORIZATION + COMMONSENSE REASONING

- **PIQA** (Bisk et al., 2020): This is a multi-choice physical commonsense reasoning and a corresponding benchmark dataset. PIQA was designed to investigate the physical knowledge of existing models.

- **WinoGrande** (Sakaguchi et al., 2019): This is a collection of 44k multi-choice problems, inspired by Winograd Schema Challenge (Levesque, Davis, and Morgenstern 2011), but adjusted to improve the scale and robustness against the dataset-specific bias. Formulated as a fill-in-a-blank task with binary options, the goal is to choose the right option for a given sentence which requires commonsense reasoning.

- **ARC** (Clark et al., 2018): This is a subset of ARC dataset with 2,590 "hard" questions (those that both a retrieval and a co-occurrence method fail to answer correctly). The ARC dataset contains text-only, English language exam multi-choice questions that span several grade levels as indicated in the files.

### C.1.4 OTHERS

- **LAMBADA** (Paperno et al., 2016): This is a dataset to evaluate the capabilities of computational models for text understanding by means of a word prediction task. LAMBADA is a collection of narrative passages sharing the characteristic that human subjects are able to

guess their last word if they are exposed to the whole passage, but not if they only see the last sentence preceding the target word.

## C.2 EXPERIMENTAL SETUP ON MOBILE

We evaluated the model run time latency on mobile. For the models to evaluate, we tested: (1) the original Llama2-7b[4] and (2) a simplified version of SHARP ($T_{next}$) where we removed the LoRA parameters. We only store the reference layers, and call those layers multiple times for the target layers in model forwarding.

We first exported the models using ExecuTorch v0.3.0[5] with the same export configurations (using kv cache, using 8-bit dynamic quantization and 4-bit weight quantization[6], for XNNPack backend). We tested the model loading and initialization time, as well as the model forwarding time using a benchmark app[7] on Xcode (version 16.0). We ran Xcode on a MacBook Air with Apple M2 chip, 16GB memory with MacOS Sonoma 14.6.1, and wire-connected it to an iPhone 16 Pro with iOS 18.1 with 8GB memory, and ran the benchmark app on the phone.

# D REBUTTAL MATERIAL

## D.1 DETAILS ABOUT STRUCTURAL PRUNING BASELINES

We evaluate the following baseline in the rebuttal:

- LayerPruning (Gromov et al., 2024), which calculates the angle distance between different layers, and then prunes a group of consecutive layers in the model such that the first and last layers in these continuous layers have small angle distance. For LLaMA2-7B, we prune the MLP layers of the 17th to 30th layers as recommended in original paper. We keep the number of removing layers the same for fair comparison and use default settings.

- LayerPruning-Adapted, similar to LayerPruning, but uses the same replacement strategy $T_{next}$ (replacing layer $2t$ with layer $2t-1$ for $t$ in [2,15]) as mentioned in Table 1 in the main paper. We also use the same LoRA ranks in LayerPruning-Adapted as SHARP. Notably, this baseline is equivalent to directly pruning the corresponding layers in SHARP, rather than reusing them as SHARP.

- LLM-Pruner (Ma et al., 2023), another advanced structural pruning algorithm. It uses gradient information to divide the parameters into several groups based on their relevance, and then prune the coupled parameters that contribute minimally to its overall performance. Here for fair comparison, we also let LLM-Pruner remove about 50% of the MLP layers except the first two and last one layers, and use the same LoRA ranks for recovering. Besides, we find that LLM-Pruner is not that stable in the in-distribution recovery tasks, even when we try the default LoRA rank and prune both the self-attention layer and MLPs together. So we report the best result in the entire finetuning process before it becomes unstable.

Here we ensure the amount of LoRA recovery components are the same for fair comparison of different structural pruning baselines.

---

[4]https://huggingface.co/meta-llama/Llama-2-7b
[5]https://github.com/pytorch/executorch
[6]https://github.com/pytorch/executorch/blob/main/examples/models/llama2/export_llama_lib.py
[7]https://github.com/pytorch/executorch/tree/main/extension/apple/Benchmark

Table 9: The downstream performance of the models utilizing 4-bit quantization. (W.G.: Wino-Grande, Med.: MedMCQA, Com.: ComSenQA)

| Models | SciQ | BoolQ | PIQA | ARC-E | ARC-C | W.G. | Med. | |
|--------|------|-------|------|-------|-------|------|------|---|
| Direct Sharing | 22.7 | 49.1 | 56.7 | 26.8 | 22.0 | 50.6 | 22.4 | |
| SHARP (w/o finetuning) | 87.8 | 66.1 | 63.0 | 54.8 | 26.8 | 57.8 | 30.5 | |
| SHARP | 92.6 | 76.0 | 72.6 | 65.8 | 34.7 | 62.4 | 31.4 | |
| **SHARP + 4-bit quan.** | 92.1 | 74.7 | 71.9 | 65.1 | 35.0 | 62.2 | 30.7 | |
| | BBH | GSM8k | MAQ. | MUT. | LAM. | Com. | QA4MRE | Avg. |
| Direct Sharing | 0.0 | 0.0 | 20.7 | 57.5 | 8.1 | 20.0 | 16.6 | 26.7 |
| SHARP (w/o finetuning) | 24.8 | 1.6 | 22.1 | 59.8 | 35.0 | 20.0 | 30.2 | 41.5 |
| SHARP | 29.8 | 3.6 | 24.7 | 68.3 | 58.3 | 44.1 | 38.3 | 50.2 |
| **SHARP + 4-bit quan.** | 27.9 | 2.8 | 24.1 | 68.1 | 55.6 | 37.4 | 39.3 | 49.1 |

