# OpenReview forum: "SHARP: Accelerating Language Model Inference by SHaring Adjacent layers with Recovery Parameters"
_ICLR.cc/2025/Conference — Submitted to ICLR 2025_

### Official Review · Reviewer_ihG5 · 2024-10-29

**Soundness:** 3
**Presentation:** 3
**Contribution:** 2
**Rating:** 5
**Confidence:** 4

**Summary:**

This paper explores the approach of utilizing an adjacent layer-sharing strategy to compress LLMs by sharing parameters between neighboring layers. The primary motivation behind this approach is based on the observation that the output features of adjacent layers are significantly similar, which suggests the potential for parameter sharing to achieve more efficient communication and inference. The specific method used involves sharing parameters between adjacent layers and introducing a layer parameter tuning mechanism using a LoRA module. This two-part process initially focuses on minimizing the L2 loss of output between adjacent layers followed by a fine-tuning phase. The results are compared against direct parameter sharing without fine-tuning to highlight the benefits of the introduced method.

**Strengths:**

1. **Innovative Approach:** The methodology combines parameter sharing with fine-tuning using a LoRA module, presenting a new avenue in model compression that could potentially conserve computational resources while maintaining model accuracy.
2. **Efficiency Gains:** The speed-up achieved through this method is notable, providing a practical solution for scenarios where faster inference is crucial, such as in mobile or edge computing environments.

**Weaknesses:**

1. **Performance in Downstream Tasks:** Despite the improvement in speed, the performance loss in downstream tasks is still significant, raising concerns regarding the practical applicability of the SHARP method in real-world scenarios where maintaining high levels of accuracy is paramount.

2. **Ambiguity in Communication Definition:** The article does not clearly define what is meant by "communication" in the context of parameter sharing. It is essential to clarify whether it refers to internal model communication, information flow between layers, or external communications between distributed components.

3. **Lack of Comparative Analysis:** The authors failed to include comparisons with alternate model compression techniques, such as LLM pruning, despite they also attempt to drop the LLM parameters while retaining performance. Such a comparison could have provided a more comprehensive understanding of how this new method stacks up against traditional approaches in terms of parameter reduction, tuning overhead, and performance impact.

Overall, while the adjacent layer-sharing strategy introduces a promising direction for LLM compression by leveraging the similarity in output features of neighboring layers, the practical limitations noted in performance impact on downstream tasks and lack of broader comparisons with established techniques present significant areas for improvement. Addressing these issues and providing a clearer exposition of the methodology and its implications could make this approach more robust and widely applicable in the field.

**Questions:**

Please see the weakness part

---

> ### Author Response · Authors · 2024-11-23
> **Response to Reviewer ihG5**
>
> Thank you for your recognition of our paper and your constructive feedback. We have responded to your concerns and will revise our paper based on the discussions. We would also appreciate it if you could let us know if our response addresses your concerns.
>
>
> > **Q1**: Despite the improvement in speed, the performance loss in downstream tasks is still significant, raising concerns regarding the practical applicability of the SHARP method in real-world scenarios where maintaining high levels of accuracy is paramount.
>
> **A1**: Please refer to ‘All-A3’ in the ‘Reply to all the reviewers’ part. Besides, in ‘All-A2’ we also show that this may be a general problem for all the structural pruning methods, while SHARP has achieved better recovery performance over other structural pruning baselines consistently.
>
> > **Q2**: Ambiguity in Communication Definition: The article does not clearly define what is meant by "communication" in the context of parameter sharing. It is essential to clarify whether it refers to internal model communication, information flow between layers, or external communications between distributed components.
>
> **A2**: Thanks for pointing out the definition that may be confusing. In our paper, ‘communication’ mainly refers to loading the model weights from low-speed, high-capacity memory to high-speed, low-capacity computation caches. For example, on mobile devices, ‘communication time’ denotes the loading time of model weights from memory to RAM/DRAM; while on server, it can denote the loading time from CPU memory to CUDA device (like ‘.to(device)’ process). We will add this in the revised paper.
>
> > **Q3**: The authors failed to include comparisons with alternate model compression techniques, such as LLM pruning, despite they also attempt to drop the LLM parameters while retaining performance. Such a comparison could have provided a more comprehensive understanding of how this new method stacks up against traditional approaches in terms of parameter reduction, tuning overhead, and performance impact.
>
> **A3**: Thanks for your constructive suggestions, we try three more advanced structural pruning methods: LayerPruning, LayerPruning-Adapted, and LLM-Pruner, and show that SHARP performs consistently better than them. Please refer to ‘All-A2’ in the ‘Reply to all reviewers’ part for the details.

---

> > ### Author Response · Authors · 2024-11-28
> >
> > Dear reviewer,
> >
> > We sincerely thank your constructive comments to make our paper better. If you have further concern, don't hesitate to let us know, and we value the opportunity for addressing any remaining issues in the discussion stage. And if you find our response address your concerns, we would greatly appreciate your support and a higher score for our paper. Thanks!

---

### Official Review · Reviewer_tUXM · 2024-11-03

**Soundness:** 3
**Presentation:** 3
**Contribution:** 2
**Rating:** 5
**Confidence:** 4

**Summary:**

This paper propose a layer sharing approach with additional low-cost finetuning to regaun the performance. After sharing the weights in the adjacent layers in the pretrained model, a two-stage lora-based finetuning is proposed, with one stage minimizing single layer output difference and the second stage recovering full model performance.

**Strengths:**

This paper provides a solid study on ways to regain model performance after layer merging. Experiments are conducted on both the ratio/location of layer merging and the ways to further finetune the model and regain performance. The paper is overall well-written and easy to follow. From the novelty perspective, layer sharing appears to be a new way of compressing pretrained large models, and the paper successfully show the runtime speed up on real devices, proving the proposed method as a promising research direction.

**Weaknesses:**

The main weakness of this work is two-fold, one is on the performance loss of the proposed method, and second is on the generalizability of the proposed finetuning method.
1. Although the proposed method show promising recovery performance on the PPL of pretraining dataset (Tab 2 and 3), we do observe significant performance drop on downstream tasks even after finetuning. This leaves me doubt if there's potential overfitting in the finetuning process, so that the performance on seen datasets are recovered but that on unseen tasks are not. A cross validation of finetuning datasets would be helpful on analyzing this issue. Additional techniques may need to be proposed to tackle overfitting.
2. Although this paper sets its background as enabling layer sharing across adjacent layers, the majority of the method is focused on the two-stage finetuning to regain performance. Finetuning is typically applied in all model compression settings, not limited to layer sharing. The porposed 2-stage finetuning may also be useful on other model compression techniques. More clarifications on the contribution is needed, as whether the 2-stage finetuning is specifically proposed for the layer sharing task, or is it borrowed from previous techniques.

**Questions:**

1. How would the model behave if the finetuning data and the evaluation task does not fully match? For example, will fineutning on GPT4-Alpaca only helps regain the performance on Arxiv-math?
2. Is the proposed 2-stage finetuning scheme limited to layer sharing? Or could it be used for other model compression methods?

---

> ### Author Response · Authors · 2024-11-23
> **Response to Reviewer tUXM**
>
> Thank you for your recognition of our paper and your constructive feedback. We have responded to your concerns and will revise our paper based on the discussions. We would also appreciate it if you could let us know if our response addresses your concerns.
>
> > **Q1**:  ... we do observe significant performance drop on downstream tasks even after finetuning.
>
> **A1**: Please refer to ‘All-A3’ in the ‘Reply to all the reviewers’ part. Besides, in ‘All-A2’ we also show that this may be a general problem for all the structural pruning methods, while SHARP has achieved better recovery performance over other structural pruning baselines consistently.
>
>
> > **Q2**: This leaves me doubt if there's potential overfitting in finetuning process, so that performance on seen datasets are recovered but that on unseen tasks are not … How would the model behave if the finetuning data and the evaluation task does not fully match? For example, will fineutning on GPT4-Alpaca only helps regain the performance on Arxiv-math?
>
> **A2**:
> Thanks for pointing out this concern. We conduct the cross-validation experiments you suggested and show the results in Table R3-1 below. We claim that **our SHARP algorithm doesn’t have a significant overfitting phenomenon on the recovering dataset.**
>
> **Table R3-1**: Explore if SHARP overfits the finetuning data used by SHARP for recovering performance. We consider the in-distribution task, and focus on the model that is only finetuned on the GPT4-Alpaca. ''using in-distribution data'' means choosing the training data that is from the same dataset as the test task
>
> ||Arxiv-math|DialogSum|GPT4-Alpaca|Dolly|OpenOrca|
> |---------------|---------------|-------|------------|----|----|
> Direct Sharing| 2171.3 | 801.7 | 20662.1 | 7221.7 | 12108.5 |
> SHARP (w/o finetuning) using GPT4-Alpaca |7.4 | 6.4| 4.2| 8.5 | 10.1
> SHARP (w/o finetuning) using in-distribution data| 4.8 | 5.3 | 4.2 | 7.2 | 8.4 |
> SHARP using GPT4-Alpaca| 4.6|4.7| 2.8 | 5.9 | 6.6
> SHARP using in-distribution data| 3.2 | 3.8 | 2.8 | 4.7 | 4.3 |
>
> Here we consider utilizing SHARP with only GPT4-Alpaca data (for both the SLW and SFT stages), and then evaluate the perplexity on different tasks. We can see that
>
> (1) SHARP (with or w/o finetuning) can consistently recover the perplexity on every task, rather than just recover model performance on related tasks like Arxiv-math, supporting that **SHARP is not overfitting to the recovery dataset**.
>
> (2) Especially, we observe that after the Single-Layer-Warmup (SLW) stage on GPT4-Alpaca, the model perplexity has recovered a lot compared to the vanilla Direct Sharing baseline.
>
> (3) We can also find that for the complete 2-step SHARP, the gap between using in-distribution data (i.e., the standard settings in Table 2 in the main paper, where the training data is from the same dataset as the test task) and GPT4-Alpaca data is not that large, which implicit that our method should have a good generalization in utilizing different recovering data.
>
> Actually, to achieve better perplexity results, we can merge the data from all 5 datasets (While In our paper, we just recover the models on each task one by one for clearer illustration).
> Thanks again for mentioning this, and we will include this useful ablation study in the revised version.
>
> > **Q3**: …The porposed 2-stage finetuning may also be useful on other model compression techniques. More clarifications on the contribution is needed / Is the proposed 2-stage finetuning scheme limited to layer sharing? Or could it be used for other model compression methods?
>
> **A3**:
> Thanks for mentioning this, we claim that:
> 1. **SFT is commonly used in structural pruning methods.** As you mentioned, SFT is widely used in the ML community, including structural pruning approaches like LayerPruning[1] nad LLM-Pruner[2].
> 2. **Things like the SLW sound not that novel, but as far as we know, we are the first work to use them in model compression**. Actually, SLW just uses the simple L2 regression loss (Line 240-247), and this is a standard choice for fitting something like ‘student-teacher’ settings, where we see the reference layer + LoRA components as a ‘student’ while the target layer as a ‘teacher’.
> However, we are the first work to try to fit the adjacent layers to save parameters and accelerate inference, therefore as far as we know, we first try to apply this SLW stage to model compression context. If you note we are missing some related works about it, we will also appreciate it very much and will add the citation. We will also add these to revised paper for clearer contribution illustration.
>
>
> [1] Gromov, Andrey, Kushal Tirumala, Hassan Shapourian, Paolo Glorioso, and Daniel A. Roberts. "The unreasonable ineffectiveness of the deeper layers." arXiv preprint arXiv:2403.17887 (2024).
>
> [2] Ma, Xinyin, Gongfan Fang, and Xinchao Wang. "Llm-pruner: On the structural pruning of large language models." Advances in neural information processing systems 36 (2023): 21702-21720.

---

> > ### Author Response · Authors · 2024-11-28
> >
> > Dear reviewer,
> >
> > We sincerely thank your constructive comments to make our paper better. If you have further concern, don't hesitate to let us know, and we value the opportunity for addressing any remaining issues in the discussion stage. And if you find our response address your concerns, we would greatly appreciate your support and a higher score for our paper. Thanks!

---

### Official Review · Reviewer_WpoQ · 2024-11-03

**Soundness:** 3
**Presentation:** 2
**Contribution:** 2
**Rating:** 3
**Confidence:** 3

**Summary:**

This paper proposes a method to reduce model memory usage and increase inference speed by sharing parameters between adjacent layers and introducing low-rank recovery parameters to maintain performance. First, Single Layer Warmup (SLW) replaces a pre-trained layer by adding a LoRA adapter to the previous layer's weights. To warm up the LoRA adapter initially, it is trained to mimic the output of the original layer. Second, Supervised Fine-tuning (SFT) fine-tunes all LoRA adapters across layers to preserve model performance. This approach significantly improves performance compared to Direct Sharing and achieves similar performance to the original model while reducing model weights by approximately 62%.

**Strengths:**

The advantage of this paper is that it can reduce model parameters by more than 50% while maintaining similar performance to the baseline algorithm in some tasks. Especially, while full fine-tuning requires a lot of data, the initial warm-up using the L2 norm of the output of the original layer can be done in parallel, which helps improve performance. If the proposed algorithm still performs well after applying quantization to compress the model, it could further reduce the model weights in an orthogonal way to quantization.

**Weaknesses:**

The main drawback of this algorithm is that it is not beneficial for all tasks. For instance, in Table 6, while SHARP improves accuracy over Direct Sharing and even surpasses the original on the CommonsenseQA task, its accuracy significantly degrades compared to the original across most tasks, raising doubts about its overall utility. Of course, reducing the model size by half could lead to an accuracy drop, but I’m still curious whether the compressed model performs better than an originally smaller model. For example, while it might be possible to compress the LLaMA2-13B model down to 7B, I’m concerned that its accuracy could be lower than that of the original LLaMA2-7B model, which does not require a complex compression process.

The training process also has a complexity issue, particularly with applying SLW using the L2 norm. Since the paper highlights potential problems with using SFT alone, modifying the model structure effectively may be challenging. This concern is especially relevant for models more complex than Llama-2-7B, which was used in the study. If SLW fails, the model may not be optimized effectively. Unlike MobileLLM, which extends an existing model, this paper’s approach removes certain layers from a pre-trained model, introducing the risk that training may become unstable if SLW fails.

**Questions:**

1. Is the accuracy of a large model reduced by SHARP higher than that of a smaller model with an equivalent number of parameters? For instance, if the LLaMA2-13B model is reduced by 7B parameters using SHARP, does this new model outperform the existing LLaMA2-7B in accuracy? If so, it would effectively demonstrate the utility of SHARP (and I'm willing to increase the score if this question is answered).

2. In Table 2, it is described that the comparison is between the model after only SLW, only SFT, and the SHARP algorithm, but there seem to be no results for when only Stage 2 was applied.

3. I am curious whether this approach yields similar results when applied to larger models and when applied to recently released models like Llama-3.2.

4. If the model is BF16, the model size would be reduced to 1/4 with 4-bit quantization. I am curious about the results of applying the idea proposed in the paper after quantization.

5. Wouldn't attaching LoRA adapters to the entire model lead to even better performance?

---

> ### Author Response · Authors · 2024-11-23
> **Response to Reviewer WpoQ**
>
> Thank you for your recognition of our paper together with your valuable comments and suggestions. We will revise our paper according to your comments. We respond to your questions below and would appreciate it if you could let us know if our response addresses your concerns.
>
> > **Q1**: ...in Table 6, while SHARP improves accuracy over Direct Sharing and even surpasses the original on the CommonsenseQA task, its accuracy significantly degrades compared to the original across most tasks, raising doubts about its overall utility.
>
> **A1**:
> Please refer to ‘All-A3’ in  ‘Reply to all reviewers’ for illustration. In short, the downstream degradation is a general problem for all the structural pruning methods (like LayerPruning[1], LLM-Pruner[2], and MobileLLM[3]), and SHARP has shown the best recovery performance on all recovery tasks in Table A2  from the ‘Reply to all reviewers’ part.
>
>
> > **Q2**: Is the accuracy of a large model reduced by SHARP higher than that of a smaller model with an equivalent number of parameters? For instance, if the LLaMA2-13B model is reduced by 7B parameters using SHARP, does this new model outperform the existing LLaMA2-7B in accuracy?
>
>
> **A2**:
> First, the main issue of the performance drops is discussed in A1 and ‘All-A3’ in ‘Reply to all reviewers’. We are sorry that due to the restriction of computation resources and time, we are unable to verify if applying SHARP on the LLaMA2-13B can outperform the existing LLaMA2-7B model on our current GPUs. However, we still have the following reasons to support the advantage of SHARP:
>
> 1. The downstream degradation is a general problem for all the structural pruning methods, and compared with advanced structural pruning methods[1,2], we have consistently better recovery performance on all tasks as shown in Table A2 in the ‘Reply to all reviewers’ part. Our analysis of performance drops (Figure 4 in the main paper) can also be helpful for enlightening future work to solve this general problem of performance drops.
>
> 2. As illustrated in the ‘OUR POSITION’ part in the ‘Reply to all reviewers’, SHARP is mainly designed for accelerating model inference by reducing the memory load overhead and is especially helpful on mobile devices which always have small RAM to load the whole model. Therefore, comparison on smaller models is more meaningful for SHARP, and in A6 we also show that SHARP is useful for the recovery of smaller models like LLaMA3.2-3B.
>
> 3. As shown in ‘All-A1’ in the ‘Reply to all reviewers’ part, we see quantization method is orthogonal to and compatible with SHARP, therefore, they can combined together for model compression for inference on mobile devices.
>
> > **Q3**: …This paper’s approach removes certain layers from a pre-trained model, introducing the risk that training may become unstable if SLW fails.
>
> **A3**:
> We claim that in general, SLW is stable. The reasons are as follows, and we will add them into the revised paper.
>
> 1. **The similarity between adjacent layers.** SLW just simply finds some recovery components to mimic the output between adjacent layers, and as shown in Figure 2 in the main paper, adjacent layers are quite similar. **This phenomenon has been verified on larger models like LLaMA2-70B or other models like Mistral-7B in the previous works [2,4]**.
>
> 2. **The simplicity of optimization loss**. The SLW stage just utilizes simple standard L2 regression loss on the output of adjacent single layers, whose optimization is empirically observed as quite stable. In the experiment, we just used about 10% of the data for the SLW stage (Line 341-343) and it’s already enough for all the layers’ SLW stage to converge.
>
> 3. **Empirical results support the claim**. The final result, like Table A2 in the ‘Reply to all reviewers’ part or Table 1 in the original paper, show SLW stage alone recovers a lot compared to vanilla Direct Sharing. And the computation cost of SLW is also significantly smaller than the SFT stage since we just do independent single-layer fittings than processing the entire large model.
>
> 4. **Robustness of recovery dataset**. The SLW stage is even robust to the choice of recovery dataset. In Table R3-1 in the reply to reviewer tUXM, we show that using GPT4-Alpaca as recovery data can still recover model perplexity, for both SLW and SFT stages. This also shows the robustness of the SLW stage.
>
>
> > **Q4**: In Table 2, it is described that the comparison is between the model after only SLW, only SFT, and the SHARP algorithm, but there seem to be no results for when only Stage 2 was applied.
>
> **A4**: In Table 3 in the original paper, we do an ablation study about different experimental settings, and ‘Supervised Fine-Tuning (10%)’ and ‘‘Supervised Fine-Tuning (100%)’ means only Stage 2 is applied. The conclusion shows that SFT is crucial for the final recovery performance, but adding Stage 1 to it can beat Stage 2 alone, especially when the rank of recovery parameters is large. (Section 3.3.3, Line 446-458)

---

> > ### Author Response · Authors · 2024-11-23
> > **continue reply**
> >
> > > **Q5**: I am curious whether this approach yields similar results when applied to larger models and when applied to recently released models like Llama-3.2.
> >
> > **A5**:
> > As illustrated in A2,  the main application scenario of SHARP is the small model and mobile devices, and we are unable to verify large models currently.
> > While for LLaMA3.2, we evaluate LLaMA3.2-3B, and the result is shown in Table R2-1 below:
> >
> >
> > **Table R2-1**: In-distribution tasks on LLaMA3.2-3B model. Here we choose T_next as the replacement strategy, which reuses about half of the MLP layers.
> >
> > ||Arxiv-math|DialogSum|GPT4-Alpaca|Dolly|OpenOrca|
> > |---------------|---------------|-------|------------|----|----|
> > Original model| 3.4 | 4.6| 3.2| 5.2 | 6.0
> > Direct Sharing| 1071.8 | 578.3 | 1000.0| 1083.7| 1690.7 |
> > SHARP (w/o finetuning) | 7.2 | 7.3| 7.1 | 12.1 | 12.5
> > **SHARP** | **4.1** | **5.2**|**4.1**| **7.6**| **6.0**
> >
> > We can see that the perplexity between the original model and SHARP is still close, supporting the generality of our method. We will add this to the revised paper.
> >
> >
> > > **Q6**: If the model is BF16, the model size would be reduced to 1/4 with 4-bit quantization. I am curious about the results of applying the idea proposed in the paper after quantization.
> >
> > **A6**: Please refer to ‘All-A1’ in the ‘Reply to all reviewers’ part.
> >
> > > **Q7**: Wouldn't attaching LoRA adapters to the entire model lead to even better performance?
> >
> > **A7**:
> > Thanks for mentioning this. In general, attaching the whole LoRA adapters to the entire model should lead to slightly better performance, and it will be more compatible with the current implementation framework of LoRA (like PEFT library). But we just apply LoRA to the replaced layer in the in-distribution tasks for clearer illustration. But we indeed are able to  attach LoRA adapters to the entire models, by first just running SLW on the LoRA that’s assigned to the target (replaced) layers (since the LoRA added to the reference layer seems should be zero if the reference layer can almost recover the target layer), and second fine-tuning all the LoRA components together in the second SFT stage.
> > Actually in the downstream recovery part (Sec. 3.4 in the original paper), we attach the LoRA adapters to the entire model as the open-instruct [5] pipeline does (Line 341-343) for convenience.
> >
> > Nevertheless, this will not result in a large difference, but mainly for convenient implementation.
> > Additionally, we show if applying LoRA to all layers or not (also just MLP layers for clarification) in Table R2-2. We can see that although more LoRA components bring difference for SLW stage (SHARP (w/o) fine-tuning), they behave the same in complete SHARP. We will add this to the revised paper.
> >
> > **Table R2-2**: Adding LoRA to the entire model. Full-LoRA means attaching the LoRA adapters to all the layers, no matter the reference layer or the target layer.
> >
> > ||Arxiv-math|DialogSum|Dolly|
> > |---------------|---------------|------------------|----|
> > SHARP (w/o finetuning) | 4.8 | 5.3  | 7.2 |
> > SHARP | 3.2 | 3.8   | 4.7 |
> > SHARP Full-LoRA (w/o finetuning) | 4.7 | 5.3  | 6.1
> > SHARP Full-LoRA  | 3.2 | 3.8   | 4.7
> >
> >
> > [1] Gromov, Andrey, Kushal Tirumala, Hassan Shapourian, Paolo Glorioso, and Daniel A. Roberts. "The unreasonable ineffectiveness of the deeper layers." arXiv preprint arXiv:2403.17887 (2024).
> >
> > [2] Ma, Xinyin, Gongfan Fang, and Xinchao Wang. "Llm-pruner: On the structural pruning of large language models." Advances in neural information processing systems 36 (2023): 21702-21720.
> >
> > [3] Liu, Zechun, Changsheng Zhao, Forrest Iandola, Chen Lai, Yuandong Tian, Igor Fedorov, Yunyang Xiong et al. "Mobilellm: Optimizing sub-billion parameter language models for on-device use cases." arXiv preprint arXiv:2402.14905 (2024).
> >
> > [4] Liu, Zichang, Jue Wang, Tri Dao, Tianyi Zhou, Binhang Yuan, Zhao Song, Anshumali Shrivastava et al. "Deja vu: Contextual sparsity for efficient llms at inference time." In International Conference on Machine Learning, pp. 22137-22176. PMLR, 2023.
> >
> > [5] Wang, Yizhong, Hamish Ivison, Pradeep Dasigi, Jack Hessel, Tushar Khot, Khyathi Chandu, David Wadden et al. "How far can camels go? exploring the state of instruction tuning on open resources." Advances in Neural Information Processing Systems 36 (2023): 74764-74786.

---

> > > ### Comment · Reviewer_WpoQ · 2024-11-28
> > > **Response to Authors**
> > >
> > > Thank you for the detailed experiments and responses. I agree with the authors that other pruning techniques also lead to accuracy drops in downstream tasks. However, I still believe that a comparison between an 'uncompressed smaller model' and the 'compressed SHARP model' is necessary. For instance, when SHARP is applied to LLaMA2-13B to create a 7B model, a decrease in accuracy might be acceptable to users (as the authors argue, the drop may be smaller compared to other techniques). However, if the compressed SHARP 7B model performs worse than the LLaMA2-7B model, users may find it unacceptable, as they could simply use the existing model without undergoing the complex compression process. Therefore, I have unfortunately decided to maintain my score.

---

> > > > ### Author Response · Authors · 2024-11-28
> > > > **Reply to reviewer**
> > > >
> > > > Thanks for your reply and constructive suggestions. However, we still want to defend ourselves by the following reasons:
> > > >
> > > > (1) We mainly focus on edge devices like mobile phones, rather than larger models like LLaMA2-13B. For example consider MobileLLM[3], their largest model just contains 350M parameters.
> > > > So **checking LLaMA2-13B is not necessary for our purpose**, and a better way is to compress a smaller model  and then run it on mobile devices, where we have checked the workability on models like LLaMA3.2-3B in A5 in the rebuttal.
> > > > More importantly, to make the model smaller enough for mobile devices, we may need to combine several model compression methods together, like combinig **pre-trained small models, quantization, and structural tuning**. They **are parallel directions, and can be used simultaneously** for obtaining a 100MB-level model that can run on mobile devices, so **we should compare them with similar approaches** for fair comparison.
> > > >
> > > >
> > > > (2) Both the previous works LayerPruning[1] and LLM-Pruner[2] don’t conduct this kind of comparison due to the restriction of data and computation cost.
> > > > Pruning LLaMA2-13B is even more expensive than pruning LLaMA2-7B in our main paper, which is unachievable at the rebuttal stage.
> > > >
> > > > (3) The value of our paper is not only the algorithm itself, but also the observation that the similarity between adjacent layers enables reusing the previous layer to predict the next (several) layers. And reusing the previous layer can be more efficient than directly pruning them as shown in Table A2 in the reply to all reviewer parts.
> > > > Besides, we also show how the capability of LLM is stored in different layers (Figure 4 in the paper), which can be helpful for guiding futural fine-grained pruning methods.
> > > > In short, we also reveal some interesting phenomenon that may enlighten futural model compression methods or interpretability works.
> > > >
> > > > We sincerely thank your feedback and hope that this can alleviate your worry.
> > > > And if these rebuttals are reasonable to you, we will really appreciate it if you reconsider your score. Thanks again for your detailed feedback to make our paper better.

---

### Official Review · Reviewer_kgWy · 2024-11-09

**Soundness:** 3
**Presentation:** 3
**Contribution:** 3
**Rating:** 6
**Confidence:** 3

**Summary:**

This paper proposes SHARP (sharing adjacent layers with recovery parameters) to accelerate LLM inference by sharing parameters across adjacent layers to reducing memory load overhead. The model performance is maintained through low-rank recovery parameters. Specifically, it employs a two-stage recovery process: SLW and SFT. Experiments results demonstrate the effectiveness in recover perplexity with a small amount of fine-tuning date while reducing the number of MLP parameters significantly. Also the inference time reduction was achieved compared to the original model on mobile devices.

**Strengths:**

+ This work follows a relative new methodology for efficient inference, i.e., adjacent layer-sharing strategy. While prior work focuses on training from scratch, this work focuses on deploying pretrained model in a resource-saving post-training way.

+ The proposed method is motivated by the robustness of LLM when replacing adjacent MLP layers and makes new observations in support of the layer-sharing strategy.

+ The two stages by SLW and SFT provide a good heuristics in layer-sharing. The work introduces low-rank weights to predict subsequent layers.

**Weaknesses:**

- The experiments are not comprehensive enough to compare with state-of-the-art efficient inference methods.

- In the latency analysis, models are simplified with 4-bit quantization to fit in iPhone. However, that is not a valid implementation since direct quantization may degrade model performance. On the other hand, it shows that weight sharing is not sufficient to support efficient inference on edge devices.

**Questions:**

Although the work improves the model performance and run time performance significantly, it only compares with the direct sharing baseline. More comprehensive comparison with state-of-the-art efficient inference methods are needed to justify the advantage of the layer sharing strategy over other categories of strategy, like those mentioned in the paper, e.g., pruning or MobileLLM. The authors could comment on the advantages of weight sharing in terms of model performance, training cost, memory resources, etc. over other state-of-the-art methods.

---

> ### Author Response · Authors · 2024-11-23
> **Response to Reviewer kgWy**
>
> Thank you for your recognition of our work and your constructive feedback to help us improve our paper. We will revise our paper based on your feedback. We detail our response below and please kindly let us know if our response addresses your concerns.
>
> > **Q1**: More comprehensive comparisons with state-of-the-art efficient inference methods are needed to justify the advantage of the layer-sharing strategy over other categories of strategy, like those mentioned in the paper, e.g., pruning or MobileLLM.
>
> **A1**: Thanks for your constructive suggestions, we try three more advanced structural pruning methods: LayerPruning, LayerPruning-Adapted, and LLM-Pruner, and show that SHARP performs consistently better than them. Please refer to ‘All-A2’ in the ‘Reply to all reviewers’ part for the details.
>
>
> > **Q2**: The authors could comment on the advantages of weight sharing in terms of model performance, training cost, memory resources, etc. over other state-of-the-art methods.
>
> **A2**:
> Here we mainly compare the structural pruning methods, because methods like quantization are orthogonal to and compatible with our method (as illustrated in A3 as follows and ‘All-A1’ in the ‘Reply to all reviewers’).
>  Compared to LayerPruning, LayerPruning-Adapted, and LLM-Pruner, we can see SHARP has better recovery capability.
>
>  Although SHARP may cost slightly more training cost and memory resources due to the existence of the Single-Layer-Warmup (SLW) stage, it costs much less than the Supervised-Fine-Tuning (SFT) stage, which is the necessary stage of all other baselines.
> The reason is that as illustrated in the paper, SLW is an independent process for different layers, and it just uses a small amount of finetuning data for convergence (like 10%). Also note that the training cost of SLW is just like fitting each single MLP layer, rather than running forward and backward processes on the whole multilayer models, the memory requirement of SLW is quite small.
> In general, SHARP is a competitive method over the previous advanced structural pruning methods.
> We will add these discussions to the revised paper.
>
>
> > **Q3**: In the latency analysis, models are simplified with 4-bit quantization to fit in the iPhone. However, that is not a valid implementation since direct quantization may degrade model performance.
>
> **A3**: In Table A1 in ‘All-A1’ of the ‘Reply to all reviewers’ part, we validate that deploying 4-bit quantization just results in about 1% downstream performance drop on average, showing that our structural pruning method SHARP is compatible with quantization method.
>
>
> > **Q4**: On the other hand, it shows that weight sharing is not sufficient to support efficient inference on edge devices.
>
> **A4**: In ‘OUR POSITION’ in the ‘Reply to all reviewers’ part, we clarify that the main purpose of SHARP is to improve the current structural pruning method and directly accelerate the model inference, instead of aggressively reducing the number of parameters as quantization. These two different directions are always focused on different key points and can be combined, as Table A1 in ‘All-A1’ also shows that SHARP is compatible with the quantization method. Using them together can efficiently accelerate the model inference on edge devices.

---

> > ### Author Response · Authors · 2024-11-28
> >
> > Dear reviewer,
> >
> > We sincerely thank your constructive comments to make our paper better. If you have further concern, don't hesitate to let us know, and we value the opportunity for addressing any remaining issues in the discussion stage. And if you find our response address your concerns, we would greatly appreciate your support and a higher score for our paper. Thanks!

---

### Author Response · Authors · 2024-11-23
**Restate the position of our paper, and reply to all reviewers for major concerns**

We sincerely appreciate all reviewers for their insightful and constructive feedback to make our paper better. We will revise our paper according to these comments.
First, we reclaim the position of our SHARP method before answering the main concerns to clarify our motivation.
And then we answer some concerns mentioned by most of the reviewers. Please refer to Appendix D in updated paper for baseline details.

> **OUR POSITION**: SHARP is a *structural-pruning*-like method to directly accelerate LLM inference by sharing parameters across adjacent layers and reducing memory load overhead.

There are diverse approaches in model compression.
Structural pruning is one of them, whose main goal is in general not to achieve the most efficient parameter reduction, but to accelerate model inference without further requirements of hardware design.
Other methods, like quantization and sparsification, can efficiently reduce the number of parameters, but they cannot always achieve the same ratio of inference speedup as the parameter savings when deployed in general hardware.
However, these methods can achieve direct optimization and is especially helpful for edge devices (like MobileLLM).

Therefore, due to the similarity, we compare our methods with the top structure pruning baselines (All-A2). Although they can not perform as well as quantization in recovering performance, they are still important directions of model compression, especially for mobile devices.
We will add these clarifications of our position in the revised paper.

>**All-A1**: Using 4-bit quantization may degrade model performance

In the Table 9 in the updated paper, **we show that 4-bit quantization will not degrade the model's performance significantly**. Compared to using SHARP alone, using SHARP and 4-bit quantization together just brings about 1% performance drop on average, which is acceptable. This also further supports that our method is orthogonal and compatible with quantization.

>**All-A2**: Can not compare with state-of-the-art efficient inference methods

Following the claim in ‘OUR POSITION’, we mainly compare SHARP to other top structure pruning baselines. We try two state-of-the-art structure pruning methods: LayerSharing and LLM-Pruner and one variation LayerSharing-Adapted, which utilize the same replacement strategy as SHARP, equivalent to SHARP except removing the target layers rather than predict them. More details are shown in Appendix D.1 in the updated paper. Results are shown in Table A2:

**Table A2**: Additional structure pruning baseline on in-distribution tasks.
||Arxiv-math|DialogSum|GPT4-Alpaca|Dolly|OpenOrca|
|---------------|---------------|-------|------------|----|----|
Direct Sharing| 2171.3 | 801.7 | 20662.1 | 7221.7 | 12108.5 |
LayerPruning  (w/o finetuning) | 87.6 | 19.2 | 86.1 | 283.1| 102.2 |
LayerPruning | 4.1 | 4.4 | 4.0 | 6.8| 5.6
LayerPruning-Adapted (w/o finetuning) | 28.3 | 19.1 | 583.0 | 86.3 | 336.7 |
*LayerPruning-Adapted*  | *3.8* | *4.3* | *3.2* | *5.8* | *5.3* |
LLM-Pruner (w/o finetuning) | 14.7 | 6.6 | 6.8 | 13.3 | 16.1
LLM-Pruner | 11.2 | 4.5 | 4.1 | 7.6 | 7.7
SHARP (w/o finetuning) | 4.8 | 5.3 | 4.2 | 7.2 | 8.4 |
**SHARP** | **3.2** | **3.8** | **2.8** | **4.7** | **4.3** |

**[Results]**: From Table A2, we can observe that
1. [SHARP vs Others]: **SHARP performs consistently better than structural pruning methods.** This shows the advantage of SHARP over structural pruning methods.
2. [SHARP vs LayerPruning-Adapted]: when number of parameters are the same, **reusing the previous layers will be more efficient than pruning**.
3. [LayerPruning-Adapted vs LayerPruning]: T_next , replaces the layers at intervals, is better than LayerPruning, which removes the consecutive layers.
4. [LLM-Pruner vs Others]: removing parameters may result in a good initial performance but be more difficult in recovering.

>**All-A3**: Still has a significant performance drop on downstream tasks

First, we admit that there are still obvious performance drops on the downstream tasks.
However, as mentioned in ‘OUR POSITION’ and All-A2, SHARP is aiming for similar improvement as structural pruning, in which field all still have the same performance degradation problem (Table A2) when the data and computation resources are limited.
Nevertheless, when given the same condition, SHARP achieves better recovery performance over structural pruning baselines consistently (Table A2), showing the advantage of our method.

On the other hand, we also do an ablation study in Figure 4 in main paper to explain the reason for the downstream performance gap in recovery, like the reasoning capability of LLM may depend on more layers and be harder to recover.

Therefore, although our method hasn’t solved the universal downstream degradation problem as in structural pruning, it has improved the current best baselines in recovering model performance and still provides useful insights for future work to solve this general problem.

---

> ### Comment · Reviewer_tUXM · 2024-11-24
> **Layer sharing and structural pruning are not equivalent**
>
> I would like to thank the author for proving this general response. The author raised a new claim in this response, stating that the proposed layer sharing is a kind of structural pruning, and "reusing the previous layers will be more efficient than directly pruning them".
>
> Besides my concern on whether raising a new claim in the rebuttal that is never mentioned in the original submission is acceptable, I cannot agree that the claim made by the author is correct.
> 1. Layer sharing and structural pruning are fundamentally different. Pruning directly removes the layer from the model, so both memory and computation are reduced. This allows the "accelerate model inference without further requirements of hardware design", as said by the author. Layer sharing, on the other hand, only reduces independent weights to be stored, but retains the full computation graph. Even for memory consumption, layer sharing may present more overhead than pruning as the locations of layers being shared need to be flagged. As for computation, the evaluation in the paper utilized "data locality", which is a special feature only available on some specific hardware platforms. Layer sharing cannot be claimed as a general accelerattion method as structural pruning.
> 2. Reusing should not be more efficient than direct pruning, no matter under what circumstances. The pruned model should have less overhead in model loading, and would have up to 2x less computation compared to the weight sharing model. The benefit of such computation reduction cannot be bridged by utilizing data locality patterns, as there's 0 cost in computing repeated layers for the pruned model.
>
> If the author want to have a fair comparison with structural pruning, the end-to-end wall-clock time should also be reported, preferably on different hardware platforms. A pruned model should be allowed to preserve more layers in a fair comparison to SHARP under the same latency, which may boost the performance of LayerPrune etc. More discussion along this line is needed.

---

> ### Author Response · Authors · 2024-11-24
> **Response to the concern - part1**
>
> We sincerely appreciate your quick response and pointing out the claims that may be confusing.
> To answer your question,
> we first restate the main advantage of SHARP for clarification, we believe that **as MobileLLM[3] does, reducing the model loading time is already a significant improvement for accelerating inference in scenarios like mobile phones, and this is the main point we always want to stick to**.
>
> **Important restatement**:
> SHARP mainly takes advantage of acceleration by directly reducing the time of loading model weights. And in scenarios like mobile devices, **This memory loading time can be much more expensive than computation.** For example, in the latency analysis (Table 7 in the original paper), we can see that the model loading time occupies 77% of the total processing time. This is also the main idea of MobileLLM[3] to reuse the previous layer (Figure 1(b) in the original paper), i.e., improve the model performance by doubling each layer, while keep the latency increase negligible since the loading time of model weights accounts for the major overhead, and reuse the previous layer almost don’t increase this part of time. Moreover, MobileLLM even shows that ways like layer reusing can doubly reduce the model initialization time, which is also longer than computation time.
>
>
>
> And then we answer your concern step by step.
>
> > **Q1**: Layer sharing and structural pruning are fundamentally different. Pruning directly removes the layer from the model, so both memory and computation are reduced.
>
> **A1**: **We mention structural pruning mainly for making it more clear that which baselines are similar to ours and we should compare with**. We admit that strictly speaking, SHARP is not the standard ‘structural pruning’ method. The standard structural pruning can not only save the parameters, but also reduce the computation cost directly. We will explain it more clearly in the revised paper.
> Our main purpose of mentioning is to find the very related pruning methods for suitable comparison, since our ‘structural-pruning-style’ method has similar parameter removing strategies, like removing layers (as LayerSharing[1]) or groups of parameters (as LLM-Pruner[2]), directly, rather than compared with other model compression methods like quantization and sparsification.
> Sorry again for the confusing statement before
>
>
> > **Q2**: …Layer sharing, on the other hand, only reduces independent weights to be stored, but retains the full computation graph….As for computation, the evaluation in the paper utilized "data locality", which is a special feature only available on some specific hardware platforms. Layer sharing cannot be claimed as a general accelerattion method as structural pruning.
>
> **A2**:
> We agree that the main advantage of SHARP is to reduce the loading time of model weights instead of computation time. However, as shown in the Important restatement, **model loading time is much more expensive (77%) than computation time (23%) on mobile devices, so method that improves it has been able to be a general and useful approach for inference acceleration**. This is also one of the main supports for MobileLLM[3], which has a very similar latency evaluation as ours.
>
> As for the computation benefit that comes from the “data locality” in SHARP, we agree that this may be related to hardware platforms (like iPhone used in the paper). However, this is not the main improvement we want to stick to. We only briefly mentioned this bonus phenomenon in the latency analysis part.
>
> > **Q3**: Even for memory consumption, layer sharing may present more overhead than pruning as the locations of layers being shared need to be flagged.
>
> **A3**:
> **We don’t agree that flagging the location of layers will be a significant cost of memory**. Here as MobileLLM[3], we at most need to mark something like the index of target layers, which just cost at most KB level of memory, while for the weight of the model can be the level of GB, which is much more than the cost of marking the location.

---

> ### Author Response · Authors · 2024-11-24
> **Response to the concern - part 2**
>
> > **Q4**: Reusing should not be more efficient than direct pruning, no matter under what circumstances. The pruned model should have less overhead in model loading, and would have up to 2x less computation compared to the weight sharing model. The benefit of such computation reduction cannot be bridged by utilizing data locality patterns, as there's 0 cost in computing repeated layers for the pruned model.
>
> **A4**:
> We agree that when the total number of parameters are the same, then compared to LayerPruning/LLM-Pruner, SHARP is not the most efficient way, since SHARP only reduces the model loading time, but LayerPruning/LLM-Pruner also saves the computation time (in practice, this part of acceleration rate will be lower than 2 times, especially for group-level pruning like LLM-Pruner). However, as in the ‘Important restatement’ part, model loading time takes the majority of latency, so they should share similar inference latency. Besides, note that these two methods have the same amount of parameters after pruning, it’s reasonable to compare them and show that our method has better performance.
>
> Nevertheless, our paper also shows a better replacement strategy than LayerPruning, as mentioned in LayerPruning-Adapted and Table 4 in paper, showing that our paper also improves the previous reusing baseline by providing more fine-grained observation of layer pruning.
>
>
> > **Q5**: If the author wants to have a fair comparison with structural pruning, the end-to-end wall-clock time should also be reported, preferably on different hardware platforms. A pruned model should be allowed to preserve more layers in a fair comparison to SHARP under the same latency, which may boost the performance of LayerPrune etc. More discussion along this line is needed.
>
> **A5**: As mentioned in A4, we show that **these baselines have the same amount of parameters after pruning, and the close latency due to the domination of model loading time**, so it’s still a fair comparison in the same pruning ratio of parameters, which is practically much easier to control compared to achieving the same latency. This is also widely used for the comparison of model compression methods.
>
> More comparison on different hardware platforms can indeed be helpful, but it’s hard to be done in this final rebuttal stage, and as mentioned above, our experiment is reasonable and shows that  SHARP has achieved better performance in the same amount of parameters. MobileLLM also mainly do the latency analysis mainly on one mobile platform, so we believe they are beneficial but not necessary experiments.
>
> In short, we thank the reviewer for mentioning these problems, and we will add them in the discussion part in the revised paper, and make our claim clearer for preventing confusion.
>
> [1] Gromov, Andrey, Kushal Tirumala, Hassan Shapourian, Paolo Glorioso, and Daniel A. Roberts. "The unreasonable ineffectiveness of the deeper layers." arXiv preprint arXiv:2403.17887 (2024).
>
> [2] Ma, Xinyin, Gongfan Fang, and Xinchao Wang. "Llm-pruner: On the structural pruning of large language models." Advances in neural information processing systems 36 (2023): 21702-21720.
>
> [3] Liu, Zechun, Changsheng Zhao, Forrest Iandola, Chen Lai, Yuandong Tian, Igor Fedorov, Yunyang Xiong et al. "Mobilellm: Optimizing sub-billion parameter language models for on-device use cases." arXiv preprint arXiv:2402.14905 (2024).

---

### Meta-Review · Area_Chair_rKYZ · 2024-12-22

**Metareview:**

After the rebuttal the reviewers still have the following concerns: The paper struggles to position its proposed method of layer sharing effectively, failing to provide a fair comparison with other post-training compression techniques, such as structural pruning, which is a more established method for reducing model size and accelerating inference. The author does not convincingly demonstrate that layer sharing outperforms pruning in terms of efficiency, nor do they show that layer-sharing models are comparable to scratch-trained models. Additionally, the claim that layer sharing can be a general acceleration method is undermined by its limited advantages over pruning, as it only reduces weight storage while maintaining full computation, with potential overhead in memory and computation. The paper lacks key evaluations, such as end-to-end wall-clock time comparisons, and does not provide sufficient evidence to justify the significance of layer sharing in the broader context of efficient deep learning.

Since 3 reviewers voted to reject it and the only positive review is not strongly arguing for acceptance, the paper will be rejected.

**Additional Comments On Reviewer Discussion:**

The reviewers and authors discussed, but in the end, some reviewers were not convinced about the above points so did not vote to accept the submission. One of the reviewers believe that the authors slightly changed the main claims of the paper in the discussion phase which is considered negative.

---

### Decision · Program_Chairs · 2025-01-22

Reject